# Generative AI and ChatGPT in School Children's Education: Evidence from a School Lesson

**Jussi S. Jauhiainen** [1,2,*] **and Agustín Garagorry Guerra** [1]

1   Department of Geography and Geology, University of Turku, 20014 Turku, Finland;
    agustin.garagorryguerra@utu.fi
2   Institute of Ecology and the Earth Sciences, University of Tartu, 50409 Tartu, Estonia
*   Correspondence: jusaja@utu.fi

**Abstract:** In 2023, the global use of generative AI, particularly ChatGPT-3.5 and -4, witnessed a significant surge, sparking discussions on its sustainable implementation across various domains, including education from primary schools to universities. However, practical testing and evaluation in school education are still relatively unexplored. This article examines the utilization of generative AI in primary school education. The study involved 110 pupils, aged 8–14 years old, studying in the 4th–6th grades across four classes in two schools. Using laptops, pupils participated in test lessons where content, text, figures, and exercises were generated and modified using generative AI, specifically ChatGPT-3.5. The results demonstrated that it was possible to use ChatGPT-3.5, as one example of generative AI, to personify learning material so that it would meet the knowledge and learning skills of pupils with different levels of knowledge. A clear majority of pupils enjoyed learning the generative AI-modified material. There is a promising potential of generative AI use in school education, supporting pupils' motivated learning and skills development. However, these tools need to be developed, refined and optimized to ensure proper adaptation and to create impactful, inclusive, and sustainable learning in schools to benefit pupils, teachers and education managers alike.

**Keywords:** generative AI; ChatGPT; school education; learning; school children; cognitive ergonomics; Spanish; history; sustainable development; inclusion

## 1. Introduction

The year 2023 marked a significant milestone in the widespread adoption of generative artificial intelligence (AI). It garnered extensive media coverage and generated substantial interest, reflected in the surge of inquiries on major global search engines. That year was not the inception of generative AI itself as the field had been under development for many years, even decades, considering the underlying machine learning techniques. However, the public release of ChatGPT-3.5, a generative pretrained transformer AI chatbot by OpenAI in November 2022, made generative AI accessible to a broader audience, requiring only internet access and a smart phone or computer. This artificial intelligence-based chatbot is capable of generating cohesive and informative human-like responses to user input [1]. Many open access tools such as ChatGPT-3.5 and DALL-E 2, the latter used to generate digital images from prompts as natural language descriptions, have made this technology widely accessible, sparking innovation in various fields (Figure 1).

In addition, the year 2023 was characterized by a remarkable proliferation of generative AI-based applications and continuous advancements, with new developments emerging on a weekly basis. Notably, generative AI expanded beyond the realm of text creation and modification, extending its capabilities to include figure design, visual art creation, music composition, and various other creative activities (Figure 1). It became particularly easy, speedy, and affordable to utilize. This accessibility led to hundreds of millions of individuals trying it out [1], many of whom became enthusiastic active users.

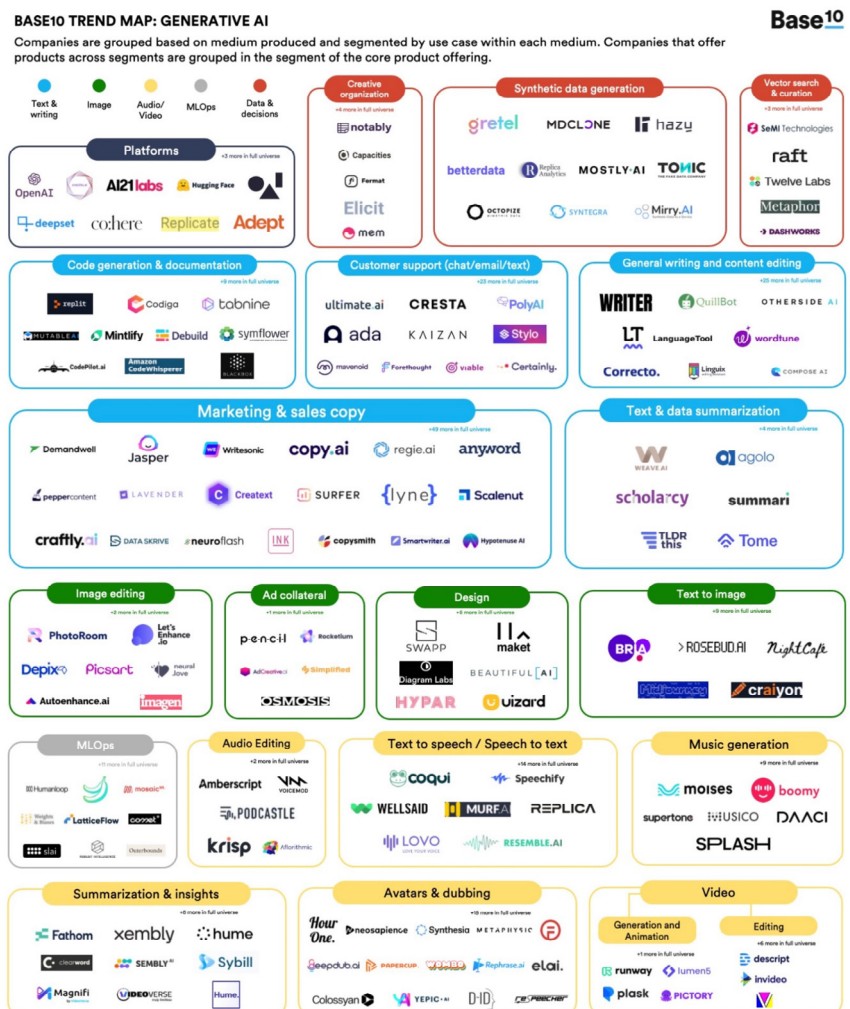

**Figure 1.** Variety of the generative AI applications in different fields as of July 2023. Source: [2].

In the coming years, the transformative impact and widespread adoption of generative AI across various fields will become more evident. While the latest developments of the Metaverse and research surrounding it have yet to deeply address sustainability concerns, generative AI holds promise in promoting sustainability efforts in increasingly digitalized and immersive environments [3]. It can play a crucial role in enhancing resource efficiency by optimizing resource utilization in various processes. This, in turn, can contribute to a more sustainable use of resources and a reduced environmental impact. Additionally, generative AI's capacity for efficient information processing can lead to improved access to valuable data, thereby aiding in people making informed decisions for sustainable practices. In particular, generative AI can support and sustain broader access to education. By providing effective personalized learning experiences, it can cater to individual needs and help bridge educational gaps, especially in underserved communities, taking into account the differences in pupils' backgrounds. This efficiency and inclusivity in education foster greater opportunities for personal and societal development, as suggested particularly by the education-related United Nations Sustainable Development Goal SDG4 [4].

Education is poised to undergo a significant transformation with the integration of generative AI. This is connected to other novelties in education such as digitalization and gamification. Earlier studies have found that digital games have the potential to increase pupils' learning engagement by enhancing their motivation, interest, and participation, also taking advantage of both written and visual learning materials [5,6]. As always, the advancement of technology and its integration into education will disrupt existing practices. Both teachers and learners need to adapt to new contexts, with their benefits

and challenges. As the Assistant Director-General for Education of UNESCO (The United Nations Educational, Scientific and Cultural Organization), Stefania Giannini noted "Over the course of my career, I have witnessed at least four digital revolutions: the advent and proliferation of personal computers; the expansion of the internet and search; the rise and influence of social media; and the growing ubiquity of mobile computing and connectivity" ... "we have, in the past several months, awoken to find ourselves abruptly entering yet another digital revolution—one which may make the others look minor by comparison. This is the AI revolution." [7].

The fundamentals of learning, such as reading, writing, and numeracy, remain essential, so generative AI cannot replace the need for acquiring these foundational skills. However, when appropriately developed and utilized, generative AI can serve as a powerful tool to support learning among learners and teachers, at all educational levels, from preschool to university.

Professionals predict that generative AI will have an impact on education, potentially also influencing the role of teachers [8]. It can facilitate teaching and learning tasks, making them more efficient and faster, but it is crucial to maintain an understanding of the processes and ensure the accuracy of the information being generated. Proficiency in critical thinking, fact checking, and comprehension remains indispensable in the era of generative AI to take full advantage of it in learning environments and teaching curricula [9]. Stefania Giannini (2023, p. 3) notes "we must preserve and safeguard the diversity of our knowledge systems and develop AI technologies in ways that will protect and expand our rich knowledge commons." [7]. Giannini and many scholars argue that the impact of generative AI in education will be at least comparable to the transformative influence of the internet and search engines.

Although there has been extensive discussion surrounding the application of generative AI in education, as of the summer of 2023, its systematic implementation in school education has been limited. This limitation pertains to its use within normal school curriculum contents and its testing with pupils in the real learning context of school classes. However, smaller tests have been conducted using school pupils and these have been analyzed by scholars, e.g., [10–16]. Most research about AI and education refer to the period before generative AI with natural language processing became an openly accessible and rather easy-to-use tool. Therefore, the key topics connecting AI and education include convolutional neural networks, artificial intelligence in general, deep learning, recurrent neural networks, learning models and data analysis, i.e., rather technical elements of AI [17].

This article presents a study focused on the utilization of generative AI in school education, aiming to contribute to the understanding and exploration of its potential in educational settings, and by doing so, contributing to SDG4. The case study here examines the design of a school lesson using generative AI, encompassing text, figures, and exercises, as well as the evaluation of pupils' learning through its implementation. Specifically, the case focuses on a Social Sciences (History) curriculum lesson designed for 4th–6th-grade pupils. In July 2023, a total of 110 pupils of 8–14 years of age in four different classes participated in and completed this lesson, and their learning experiences were assessed both quantitatively and qualitatively. The study was conducted in two schools in Montevideo, Uruguay. The article also explores the process of creating a school lesson using generative AI and testing its application in a school environment. ChatGPT-3.5 was the main application utilized for the generation of the test and related material.

The research questions were as follows: Can generative AI be used to provide personified learning material for pupils with basic, intermediate, or advanced knowledge about the topic in a school lesson? Does the time length spent on learning generative AI-modified material differ among pupils from diverse backgrounds? What perceptions do pupils have of learning with generative AI-modified material? Answering these research questions indicates the current state of the art in using generative AI to provide personified learning material for a diversity of pupils. Providing learning materials that fit the needs of a diverse

range of pupils and motive them to learn is one important prerequisite for the long-term inclusiveness and sustainability of education.

Following the introduction, this article explores the role of generative AI in school education. The discussion encompasses the conceptual framework of generative AI and highlights the practical experiences gained thus far in utilizing this technology within educational settings. A particular emphasis is placed on ChatGPT-3.5, a widely used generative AI application. The subsequent section focuses on the materials and methods employed in this study. Descriptive statistics and cross tables were predominantly used as the analytical methods to analyze the practices and responses of 110 pupils. The article proceeds to present the findings of the empirical study, including the results from the introductory, content, and self-evaluation sections of the test. Finally, the article concludes with key insights and suggestions for future research in this area. The test was conducted using the Digileac platform (see https://sites.utu.fi/digileac) application, accessed on 15 September 2023 [18].

## 2. Generative AI and the ChatGPT in School Education

Education is a fundamental activity for individuals, communities and societies. Contemporary education faces numerous opportunities and challenges, from infrequent to frequent disruptions. Many opportunities arise with the advancement of digitalization. The internet provides an expanding amount of information and knowledge to school children and teachers who have an internet connection, the ability to access these information sources, and the capacity to understand this information and process it further to new knowledge. Nevertheless, despite the shrinking digital divides from year to year, as of 2023, 35.4% of the world population had not yet been connected to the internet [19].

Apart from digital divides, there are also other major challenges that encompass the smooth implementation of education. These include global-level pandemics, such as the COVID-19 outbreak, the increased occurrence of extreme natural disasters due to climate change, recurring instances of war in numerous countries, strikes, and political conflicts, even within welfare states, prevailing poverty conditions preventing hundreds of millions of children from attending school education, particularly in less-developed countries, and the lack of updated and suitable learning materials for every child. Education is not yet globally inclusive, which creates challenges for the long-term sustainability of societies.

In the face of such challenging circumstances, there is a growing need for a responsive education system that ensures the access and continuity of educational curricula for every child, irrespective of their socioeconomic status, language, or cultural background. This article argues that generative AI can play a significant role by serving as a mediator between educational institutions, teachers and pupils, providing educational support anytime and anywhere. Properly implemented generative AI allows teachers to better focus on education opportunities and challenges, while pupils can receive guidance and support suitable to their knowledge backgrounds and levels. This would positively affect the understanding and practice of environmental, economic, and social sustainability around the world [20].

Furthermore, generative AI has the capacity to facilitate education in multiple languages, enabling the adaptation of curricula to each pupil's native language once the models have reached those levels [20]. While internet access facilitates the seamless utilization of generative AI in learning, it is not yet universally accessible. Generative AI requires globally significant investment in hardware and software, as well as ongoing maintenance and support [21]. However, it is possible to leverage its benefits in offline contexts as well through thoughtful design, and reduce the main infrastructure costs substantially. In such scenarios, a basic smartphone would suffice to access a high-quality learning environment tailored to meet the demands of the school curriculum, as well as the individual and collective needs of pupils, as indicated later in this article with the case of using the Digileac platform for the test lesson [18].

*2.1. Background of Generative AI in Education: Large Language Models (LLM) and Pre-Trained Language Models (PLM)*

The concept of generative AI refers to a type of artificial intelligence built using a machine learning system commonly known as generative models. The main characteristic of these models is their capability to generate new content. There are many types of generative AI models, and they can generate a wide variety of content. The most well-known ones generate text, images, or music. Depending on the specific case, these models will be trained on different types of data, such as text format, image format, or musical data, to compose new content from their initial libraries. Large Language Models (LLMs) have advanced substantially in natural language processing (NLP) in recent years. LLMs have been trained on massive amounts of text data so that they can generate human-like text, answer questions, and complete other language-related tasks with high accuracy [9].

As mentioned, ChatGPT is one example of LLMs. GPT stands for the combination of three concepts: generative, pre-trained, and transformer. A pre-trained model with a transformer architecture is able to generate content using Large Language Models [22]. The most commonly known format is ChatGPT, which allows the interaction with this generative AI in a chat environment, simulating almost-real conversational situations between humans [23–25].

To understand GPT technology, it is important to clarify the significance of Pre-trained Language Models (PLMs) as machine learning models that undergo training for a specific task using a vast corpus of text. The PLM is first trained on a large dataset and then it is fine-tuned on a specific task. During this training process, the model develops an understanding of statistical patterns in the provided text libraries, including word co-occurrence, sentence structures, and grammar. Subsequently, in the fine-tuning stage, the PLM assimilates how to effectively apply the acquired knowledge to accomplish a given task [26].

Various approaches are employed to train and fine-tune models, aiming to enhance their performance and ensure more reliable outputs. These approaches include methods such as semi-supervised learning and reinforced learning from human feedback (RLHF). In RLHF, human intervention is incorporated at various stages to validate or annotate incorrect outputs. This process allows the model to assign greater importance to accurate results and penalize mistakes, subsequently refining its performance through fine-tuning. Although manual fine-tuning requires significant human resources and time, its impact on model performance often exceeds that of approaches without RLHF [27].

LLM can be used to create educational content, improve pupils' engagement and interaction, and personalize their learning experiences. ChatGPT is an example of a trained LLM that has been trained on a vast corpus of text data obtained from the internet. Based on the architecture of GPT-3.5, ChatGPT utilizes neural networks in machine learning to comprehend and generate text. During the training process, the model learns patterns and relationships between tokens, which involves the tokenization and understanding of co-occurrence patterns. This encoding process, also known as embedding, allows the model to estimate the probability of predicting the next word [27]. What distinguishes ChatGPT is its ability to perform complex embeddings by incorporating contextual cues and employing self-attention mechanisms, enhancing its understanding and contextual comprehension of the text. This advanced understanding enables the model to grasp key concepts and the overall context of the given text. With an impressive 175 billion parameters, GPT-3.5 is considered an LLM due to its extensive training on vast amounts of text data [22–24]. In the end, in the school context, ChatGPT uses algorithms to generate new text and responses similar to what a teacher or a pupil might write [28].

In addition, the conversational capabilities of NLP models make it possible to utilize many languages. However, there are still limitations of NLP technologies, particularly in relation to Low-Resource Languages (LRL). Research into NLP has primarily focused on a subset of approximately 20 languages out of more than 6000 languages worldwide. However, by 2022 BLOOM had already been developed, a transparently trained multilingual language model covering 46 natural languages [29]. Nevertheless, LRLs have limited

coverage if the goal is to use generative AI in everyone's mother tongue around the world. LRLs face challenges due to the scarcity of digital content available for processing, hindering the development of robust NLP models. However, the development of generative AI and related models facilitates the integration of diverse digital applications into a unified framework or application. This offers possibilities for the global but contextualized scaling of these applications for educational purposes [30,31].

In fact, the increase in the accuracy of generative models opens the door to cross language barriers and provides high-quality content in specific languages. Currently, this area of AI is developing extremely rapidly due to several factors, including the widespread use of NLP and the continuous growth of massive datasets in open-source communities. ChatGPT-4 is showing advanced levels of accuracy in machine translation. It is expected that in the near future, NLP models will break down the language barrier, closing the gap in content accessibility. So far, GPT-4, utilizing three-shot accuracy on machine-intelligence learning optimization across languages, has been proven to increase performance across languages. For example, in English, the accuracy increased from 70.1% to 85.5% between GPT-3.5 and GPT-4. This result is also observed in other languages, where GPT-4 exhibits levels of accuracy between 80% and 85% in many languages such as Afrikaans, French, German, Indonesian, Italian, Polish, Russian, Spanish, etc. [31].

### 2.2. Current Practices of Generative AI in Education in Schools

The context of schools presents a unique opportunity for the implementation of generative AI and GPT technologies to support learning materials and teachers in effectively engaging with diverse profiles of pupils. In today's education landscape, teachers face a tremendous demand, including significant time investments in class coordination, planning, and administrative tasks, rather than being able to focus fully on teaching.

A review of the ChatGPT uses in education tests until March 2023 has showed that almost all tests were conducted in the context of higher education and universities, rather than in compulsory or secondary school education. Furthermore, its focus was on two main aspects. Firstly, ChatGPT was used in the preparation of teaching by generating course materials, providing suggestions, and performing language translation. Secondly, it was used to generate assessment tasks for the performances of students [1]. There were also technical analyses on details related to the use of GPT for school education purposes, which found several practical outcomes such as the use of the ChatGPT-3.5 for making quizzes [32].

By leveraging LLMs and generative AI technologies, teachers can alleviate some of the burdens associated with lesson planning, personalized content creation, differentiation and personalized instruction, assessment, and personal professional development. This allows them to concentrate on addressing the needs of their pupils more effectively [1,7]. For example, it offers teachers the ability to create customized content and assessments without the need to spend extensive time searching for suitable exercises for each pupil in the class. It provides teachers with a broader range of possibilities for adapting content and proposing multiple exercise options. Furthermore, it expedites the evaluation process, enabling teachers to focus their attention on pupils who require additional support, whether they are struggling or excelling. LLMs can generate contextualized natural language texts and various types of multimedia content, combining several AI systems together such as ChatGPT and DALL-E [33].

For pupils, the implementation of generative AI offers numerous benefits that can be targeted to their ages. In primary school education, pupils can receive assistance in learning reading, vocabulary, writing, and counting from the very early stages, as well as more advanced topics like writing styles, grammar, and text interpretation [10,12,16,32,34]. LLMs can be used to make a complex and long text easier and more concise. For example, the application can offer customized feedback, analyze individual needs, and assess results. This was found out in a test in which primary school pupils practiced their vocabulary [9]. According to a comprehensive review, LLMs can generate questions and prompts that

encourage pupils to think critically about what they are reading and writing [9]. In addition, they can analyze and interpret the information presented to them, which helps pupils to understand the topics better.

In secondary school education, the use of generative AI allows for a more nuanced approach in tailoring content to suit the abilities and interests of each individual student across a wide range of subjects, from natural sciences to humanities and languages. This personalized approach, facilitated by generative AI, promotes engagement and motivation in the learning process. LLMs have the capability to generate practical problems and quizzes that aid pupils in better understanding the material they are studying, including its contextualization. For instance, generative AI empowers detailed performance tracking, enabling the delivery of personalized content that aligns with each student's individual capabilities. This personalized approach helps prevent students from experiencing anxiety or boredom during classes, which are factors that can contribute to dissatisfaction and even dropout rates within the educational system [9].

LLMs can suggest appropriate variations for each pupil, thereby fostering the development of their problem-solving skills. For instance, explanations, step-by-step solutions, and intriguing related questions can be provided to pupils. This approach enables a deeper comprehension of the underlying reasoning behind solutions and encourages pupils' analytical thinking and creativity. By leveraging generative AI effectively, pupils can advance through BLOOM's taxonomy, progressing from remembering and understanding the core learning of the curriculum to advance to applying, analyzing, and assessing complex learning, then further expanding to evaluation, and ultimately culminating in the creation of new knowledge through critical thinking in deep learning (Figure 2). A study found that, for example, ChatGPT-3 can be used to increase pupils' curiosity regarding a topic studied and strengthen their skills to create associations between topics studied [35], thus advancing them along BLOOM's taxonomy. At this early phase of development, the study concluded that a sensible suggestion was to neither outright reject nor entirely rely on generative AI in school education [16].

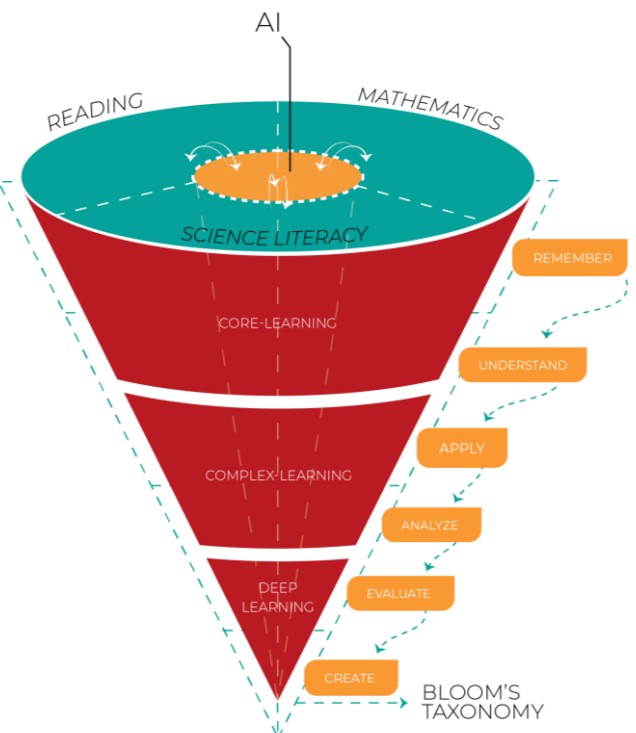

**Figure 2.** Generative AI use in education and BLOOM's taxonomy.

In another study, ChatGPT-3 was utilized to generate curiosity-inducing cues, thereby incentivizing pupils to ask more profound and thought-provoking questions. The findings demonstrated that LLMs possess significant potential in facilitating curiosity-driven learning and promoting the increased expression of curiosity among pupils [35]. Moreover, conversational AI can be employed to introduce variety to text-based exercises [36].

Pupils are more likely to remain actively involved when they feel that their unique needs are being addressed. This personalized approach also helps to optimize the learning experience by providing pupils with content that matches their current learning skill level, allowing them to progress at their own pace. In this way, the use of generative AI can enhance pupil satisfaction, promote a positive learning experience, and ultimately contribute to higher retention rates within the educational system. This motivated progress can be termed cognitive ergonomics. Ergonomics can encompass everything from the physical activities and demands of a job to how the human mind understands instructions and interfaces [36]. Cognitive ergonomics, thus, refers also to motivated cognitive interaction between the pupil and the curriculum content and how the generative AI is able to adapt content optimally for pupils' cognitive abilities [37].

Undoubtedly, while generative AI presents promising opportunities in education, it is important to acknowledge the challenges, risks, and limitations that accompany its early-stage development. These issues are not specifically addressed in the conceptual part of this article because there is already a comprehensive overview of the challenges and risks associated with the use of LLMs in education [9]. These challenges and risks encompass a range of areas, including copyright concerns, bias and fairness considerations, the potential for overreliance on the model by learners and teachers alike, the need for adequate understanding and expertise, the difficulty in differentiating between model-generated and user-generated answers, the costs associated with training and maintenance, data privacy and security concerns, the sustainability of usage, the verification of information and maintenance of integrity, the challenge of discerning genuine knowledge from unverified model output, the lack of adaptability, the absence of suitable user interfaces, and limitations in multilingual support and equitable access [9]. Obviously, GPTs can generate incorrect information since they are still far from being highly accurate. Therefore, human intervention is needed to avoid hallucinations. There are already international guidelines such as UNESCO's Recommendation on the Ethics of Artificial Intelligence [38]. However, the overall guidelines for the ethical and honest use of generative AI in school education, teaching and learning are still to be designed and approved internationally and in national and local contexts. These factors highlight the importance of addressing these concerns as generative AI evolves in the educational context.

## 3. Materials and Methods

This article presents findings from a test lesson that examined the implementation of generative AI in the context of the 4th, 5th and 6th grade pupils' Social Sciences curriculum, in particular regarding History. The primary source of data for this study comprised the responses of 110 pupils, who were 8 to 14 years old, and participated in the test lesson as part of their ongoing annual curriculum in a school class. The respondents in the 4th grade (50 pupils) were 8 to 10 years old, 5th grade (32 pupils) 11 years old, and 6th grade (28 pupils) 11 to 14 years old.

The tests were carried out in Uruguay, involving a total of four classes across two schools in Montevideo. Given that Spanish is the language of instruction, it was utilized in the test as well. One significant factor in choosing Uruguay, located in the southern hemisphere, was that during the testing period in July 2023, most schools in the northern hemisphere were closed for summer vacation. Nevertheless, subsequent tests in different languages were already planned for schools in the northern hemisphere at the time of this initial endeavor.

To implement the test lesson and the following analysis, national and university guidelines on research integrity and ethics were rigorously followed. Prior to conducting the test, following good research practices, the directors of the participating schools granted permission, and the teachers and pupils were duly informed about the testing process, including research integrity and ethical guidelines. To guarantee that all these details and research procedures were meticulously implemented, and that potential harm to the research participants would not be caused, one of the authors of the article personally observed and guided the test procedures on-site. All test materials were automatically collected and recorded through the designated test platform and stored in a secure digital framework. To ensure participants' anonymity, they answered the test without providing their names, and their identities could not identified from the results. The pupils' parents or guardians had the option to prevent their child's participation in the research. In addition, pupils took part in the test voluntarily, and they had the right the withdraw from the test at any moment or to leave questions without answers. In fact, the material only contains results from those pupils who attended the whole test lesson and answered the questions. In addition, safeguards, data practices and management followed the research code of conduct in accordance with discipline-specific principles. The conducted research is part of broader research project regarding generative AI in education, in which related research integrity and ethics procedures have been formalized and implemented administratively.

The test platform was accessed in schools using laptops. Of all the pupils, 60 used the national Ceibal program operating system. In another school, 50 pupils utilized Chromebook, which was used for regular learning activities in their respective classes. For the sake of pupil anonymity, the names of the schools located in the capital of Uruguay are not disclosed in this article.

To ensure that language did not influence the pupils' performance in the test and to align it with their mother tongue and language of instruction, Spanish was chosen as the language for this research. As of January 2023, English accounted for the majority (58.8%) of web content and is widely recognized as the most proficient language for NLP models and conversational technologies. However, Spanish is classified as a High Resource Language (HRL). It holds a significant presence on the web, representing 4.3% of website content [39], making it the third most commonly used language. This context has implications for achieving the optimal performance with ChatGPT. As it is known, in general, the model's training is based on digital material extracted from various online sources.

The analysis of the results involved employing descriptive statistics and cross-tabulation techniques to establish connections between the pupils' background information and their test performance.

*Creating the Test Material*

This article represents globally one of the initial comprehensive tests and analyses conducted to examine the implementation of generative AI in the school curriculum and evaluate its impact on pupils' learning, as well as their perception of learning, during ordinary lessons. Therefore, here we provide a detailed account of the methods employed to collect the data, as well as the variations observed within the gathered material. This aims to offer a comprehensive understanding of the test process and the diverse outcomes observed, including how much time pupils in different knowledge categories took to complete the test and what the learning results were in each knowledge category.

The first step in this study was selecting a specific topic from the school curriculum to examine the utilization and impact of generative AI on pupils' learning. A 4th- and 5th-grade Social Sciences class, which focused on the early history of humankind and particularly the life of early humans and native Americans, was an appropriate choice. For the case of 6th grade, the chosen topic was the Roman civilization. Both topics are typically covered in a 30–45 min lesson within the school curriculum and were suitable for the pupils who participated in the test.

The subsequent step involved collecting and generating foundational material for a school lesson focusing on the chosen topic. This material was designed to align with the typical content covered within a single school lesson. For the test lesson, the original reference material was sourced from worldhistory.org, and validated by the UNESCO Archives. This platform offers one of the most relevant open educational resources on history, available in multiple languages, and serves to inspire individuals to engage with cultural heritage and the teaching of history worldwide. The selected text for the 5th grade was a part from an article regarding the history of early humankind, authored by Emma Groeneveld in the World History Encyclopedia and translated into Spanish by Waldo Reboredo Arroyo [40]. For the 4th and 6th grade, a part from an article written by Teressa Kiss regarding Los Indígenas and Imperio Romano was used [41,42].

Once the original text had been validated in both English and Spanish, additional translations were performed using ChatGPT-3.5 to assess the results. Both of these generative AI models demonstrated the ability to accurately translate the content by employing the same prompt (as a written text input to ChatGPT to generate a specific useful written response), yielding outputs that closely resembled human translation. The temperature (as the parameter to control the randomness of predictions in the output text) was limited, which proved crucial in avoiding potential hallucinations (different or not realistic text) within the AI-adapted text. Notably, there was a key difference in the format of the answers produced by the developed NLP models. ChatGPT-3.5 exhibited the freedom to choose the placement of the author and link source, while ChatGPT-4 displayed consistency in adhering to the original text format provided. With this initial verification step completed, a foundational content material was established for integration into the adaptive model for generative AI usage.

The generative AI, specifically ChatGPT-3.5 and ChatGPT-4, was employed to tailor the lesson's main text (as prompt) and exercises (completion) to match each pupil's knowledge base. This was achieved by the ChatGPT-3.5's ability to modify the content to match with the pupils' previous knowledge, dividing the text and exercise material into three levels: basic, intermediate, and advanced. Consequently, a set of open-ended questions (two written questions to which pupil could freely write a more than simple one-word answer) and multiple-choice questions (two written questions and to each question a set of three predetermined answers, of which one was correct) were designed by ChatGPT-4 to suit these three groups, ensuring the appropriateness and relevance of the assessment for each pupil's level of understanding. The suitability of the text and exercises were validated by the authors of this article, following the RLHF principles.

In addition to text and exercises, the lesson material incorporated content-related illustrations, which were generated using the generative AI program (Midjourney). These figures encompassed both basic and advanced details about the topic under study. The selection of figures was tailored to cater to the varying knowledge levels of the pupils. As a result, the learning material for the lesson consisted of three sets of text, figures, and exercises representing basic, intermediate, and advanced levels of complexity. This approach aimed to effectively align with the pupils' individual learning skills and knowledge pertaining to the subject matter.

Next, to allocate the pupils into the aforementioned three groups, a preliminary step was taken before diving into the main lesson contents. Immediately after opening the application to be used for the test, prior to engaging with the main material, the pupils' presented their grades in four key school subjects. In addition, the remaining two questions gauged the pupils' self-perception of their familiarity and their interest with the topic of the lesson. Pupils indicated this before entering into the text and exercise material. Based on their mark in the related school subject of Social Sciences, each pupil was then assigned to either the basic, intermediate, or advanced group. Consequently, the material provided to each pupil was tailored to best suit their level of knowledge and learning skills.

Each group was presented with two sets of topics. Following the reading of the text and figures from the first set, the pupils were provided with two open-ended questions and two multiple-choice questions related to the content. Based on the accuracy of their responses, i.e., how correct were their answers referring to the material they had learned during the test, each pupil was assigned to either the basic, intermediate, or advanced knowledge group for the second part of the lesson. Depending on their performance, pupils could either be placed at a higher level than initially in the test, remain in their current group, or be assigned to a lower level in the second part of the lesson. This practice aimed to fine-tune pupils' learning skills and knowledge through tailored learning tasks. In the second part, pupils once again answered two open-ended questions and two multiple-choice questions based on a modified version of the text, taking into account their knowledge base from the first part of the lesson.

Finally, upon finishing their responses to the provided questions, the pupils were asked to provide their subjective perception of their learning experience during the lessons, as well as their level of satisfaction with the use of these devices and methods. Gathering feedback from the pupils regarding their learning and enjoyment was deemed crucial in the ongoing development of the learning with the help of generative AI.

## 4. Results

### 4.1. Conducting the Introductory Part of the Test Lesson

The initial phase of the test involved collecting information about the respondents. Upon opening their test devices, the pupils were first required to indicate some key demographic features, and then features regarding four key subjects (Social Sciences, Mathematics, Spanish, English) at school and their favorite subjects among these four topics.

The first question was optional, allowing respondents to indicate their gender as either boys or girls. Among all the participants, 57.3% identified themselves as boys, 40.9% identified as girls, and 1.8% chose not to answer this question. Our on-site observations confirmed that the indicated gender proportions closely matched the actual gender distribution of the pupils.

In the second question, the pupils were asked to indicate their overall school grades (ranging from poor to excellent, i.e., from 1 to 5) in four subjects: Mathematics, their mother tongue (Spanish), a foreign language (English) and Social Sciences, the latter connected to the topic that was the theme of the test. The responses varied among the pupils. The general grade of all four topics (mean 3.7; median 3.6) showed little variation among the different classes. It was 3.8 (median 3.8) in the 4th grade, 3.8 (median 3.9) in the 5th grade, and 3.5 (median 3.5) in the 6th grade (Table 1).

**Table 1.** Pupils' (110) responses to background questions (%).

|  | All Pupils | Boys | Girls | Excellent or Good Knowledge | Much or Somewhat Interest | Pupils N |
|---|---|---|---|---|---|---|
| Excellent or good school mark, average | 73.6 | 76.2 | 73.3 | 91.0 | 85.3 | 81 |
| Medium mark, average | 25.5 | 23.8 | 24.4 | 9.0 | 14.7 | 28 |
| Satisfactory or poor school mark, average | 0.9 | 0.0 | 2.2 | 0.0 | 0.0 | 1 |
| Excellent or good knowledge in topic | 30.0 | 25.3 | 35.5 | - | 55.9 | 33 |
| Medium knowledge of topic | 31.8 | 39.7 | 22.2 | - | 26.5 | 35 |
| Satisfactory or poor knowledge in topic | 38.2 | 34.9 | 42.2 | - | 17.6 | 42 |
| Much or some interest in topic | 30.0 | 31.7 | 28.6 | 57.6 | - | 34 |
| Fair interest in topic | 29.1 | 30.2 | 28.9 | 27.2 | - | 32 |
| Little or no interest in topic | 40.0 | 38.1 | 42.3 | 15.2 | - | 44 |

As regards the Social Sciences evaluation, the distribution for the whole group was the following: excellent (16.4%), good (37.3%), medium (34.5%), satisfactory (10.0%), and poor (1.8%). However, there was variation between the classes. In the 4th grade, the marks were excellent (18.0%), good (28.0%), medium (40.0%), satisfactory (12.0%), and poor (2.0%). In the 5th grade, the shares were 25.0%, 43.8%, 28.1%, 3.1%, and 0.0%. In the 6th grade they were 3.6%, 46.4%, 32.1%, 14.3%, and 3.6%. The evaluation of Social Sciences was used to place the pupils initially in the basic, intermediate or advanced knowledge groups.

In the third question, the pupils were asked to mention their favorite subject in school among the four mentioned subjects. Mathematics was the favorite for nearly half of the students and this did not vary substantially between classes: for 46.4% of pupils it was the favorite and for 17.3% pupils it was the least favorite (50.0% vs. 12.0% for the 4th grade, 40.6% vs. 12.5% for the 5th grade and 46.4% vs. 32.1% for the 6th grade).

Other school topics received the following shares: mother tongue (Spanish) was the favorite for 23.6% of all pupils vs. the least favorite for 21.8% of all pupils: 46.0% vs. 6.0% for the 4th grade, 3.1% vs. 46.9% for the 5th grade and 7.1% vs. 21.4% for the 6th grade). The foreign language (English) was perceived as the most favorite by 15.5% of pupils vs. the least favorite by 34.5%: 4.0% vs. 42.0% for the 4th grade, 18.8% vs. 21.9% for the 5th grade and 32.1% vs. 35.7% for the 6th grade. Finally, Social Sciences was on average less preferred among the pupils, as 14.5% said that it was their favorite vs. 26.4% who perceived it as their least favorite: 0.0% vs. 40.0% for the 4th grade, 37.5% vs. 18.8% for the 5th grade and 14.3% vs. 10.7% for the 6th grade.

In the fourth question, the pupils assessed their self-perception of the knowledge they possessed regarding the test topic. Specifically, it was "The history of early humankind" for the 4th-grade pupils, "Native Americans" for the 5th-grade pupils, and "The Roman civilization" for the 6th-grade pupils. The responses exhibited variation among the participating pupils. However, those having the most knowledge about the topic were also the most interested in it. More specifically, 9.1% of all pupils claimed to have excellent knowledge about the topic, while the remaining had good (20.9%), medium (31.8%), satisfactory (18.2%), and poor (20.0%) knowledge. Among the pupils in the 4th grade, these percentages were 8.0%, 26.0%, 38.0%, 24.0%, and 4.0%. Among the pupils in the 5th grade, the breakdown was as follows: 18.8%, 21.9%, 46.9%, 6.2%, and 6.2%. Among the pupils in the 6th grade, the self-perception grades were substantially lower: 0.0%, 10.7%, 3.6%, 21.4%, and 64.3% (Table 1).

The fifth background question inquired about the level of interest each pupil had in the study topic. The responses displayed significant variation, with the following breakdown: much (15.5%), some (15.5%), fair (29.0%), little (15.5%), and no interest (24.5%). For the 4th grade, the breakdown shares were 10.0%, 12.0%, 36.0%, 16.0%, and 26.0%. For the 5th grade, they were 31.3%, 21.8%, 34.4%, 12.5%, and 0.0%. For the 6th grade, these were 7.1%, 14.3%, 10.7%, 17.9%, and 50.0% (Table 1).

*4.2. Conducting the Content Part of the Test Lesson*

Following the introductory part of the test, the pupils proceeded to engage with the test lesson's learning material belonging to the Social Sciences curriculum, and specifically a relevant topic in History, as indicated earlier.

The core dataset for analysis comprised both quantitative and qualitative data centered on pupils' interaction with and reactions to the provided learning material. Their levels of engagement, attention to the material, and proficiency in comprehending it was analyzed, as evidenced by the accuracy of their responses to questions pertaining to the learning content and how much time they used to study and answer it.

The measurement of their time usage was conducted in various ways. Firstly, the amount of time dedicated to reading the provided learning material before responding to the initial questions was quantified in seconds. Subsequently, this time duration was examined in relation to the pupils' background variables. Secondly, the time consumed by each pupil to address the questions was measured. This included both open-ended

and multiple-choice questions, constituting a component of the test's time-management dimension. Thirdly, the cumulative time expended by each pupil—from their initiation of interaction with the study material to the conclusion of the final question—was gauged. This comprehensive measurement yielded insights into the overall time allocation by the pupils during their involvement with the test, thereby serving as a pivotal metric for evaluating their test performance.

The text and figures were designed to align with pupils' prior knowledge about the topic, as described above. In the first part of the test, the pupils were placed in the basic, intermediate, or advanced knowledge group based on their school grades. The text was adapted using ChatGPT-4 and verified by the authors to appropriately match the diversity of pupils' knowledge backgrounds in these three knowledge-level groups. Then, each pupil got acquainted with the text and illustrations in their respective knowledge group and responded to two open-ended questions and two multiple-choice questions regarding the learning material.

Subsequently, in the test's second phase, pupils familiarized themselves with another set of learning materials, once again tailored through the use of ChatGPT-4. This was followed by their responses to two open-ended questions and two multiple-choice questions. Based on the accuracy of their answers in the initial test phase, pupils were assigned to the corresponding higher, equivalent, or lower knowledge group for the second phase. Consequently, the learning material and questions were strategically designed to align with the pupils' proficiency levels (Figure 3).

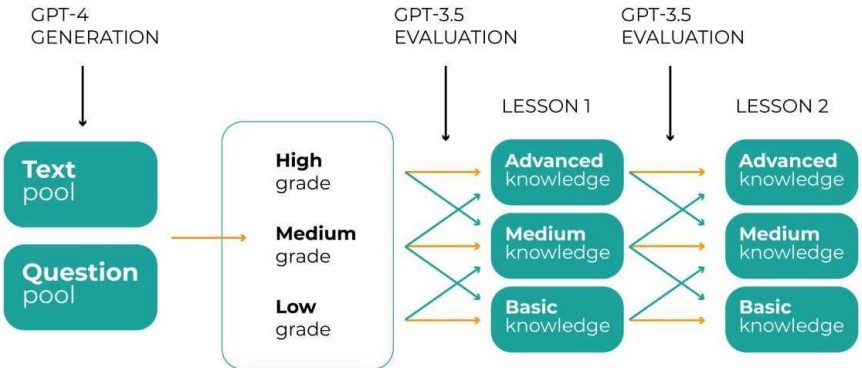

**Figure 3.** Learning-material adaptation with generative AI and pupils' learning development.

In summary, the generative AI adaptation took place in the following way (Figure 3). Each pupil engaged in a single test lesson centered around History, which was divided into two parts. Each part encompassed personalized learning material created with generative AI, categorized across three knowledge level groups. Consequently, this led to the generation of two customized texts (adapted with ChatGPT-4) and two images (previously generated using Midjourney), along with a total of four open-ended questions and four multiple-choice questions (adapted with ChatGPT-4), all designed to assess each pupil's learning. Given the diversity of pupils and the generated learning materials, descriptive statistics and cross-tabulation were employed in the analysis.

The length of a text modified by the generative AI is not a direct indicator of its complexity. The original text for the 4th grade had an average of 117 words. The AI-generated adapted text had 137 words at the basic level, 123 words at the intermediate level, and 125 words at the advanced level. For the 5th grade, the original content averaged 129 words. The AI-generated adapted text had 119 words at the basic level, 119 words at the intermediate level, and 144 words at the advanced level. For the 6th grade, the original content contained 162 words. The AI-generated adapted content had 220 words at the basic level, 195 words at the intermediate level, and 164 words at the advanced level.

As mentioned, two sets of material were provided, differentiated between basic, intermediate and advanced knowledge groups. As regards the first set of learning material, which included text and one figure, on average, 4th-grade pupils spent 92.48 s (with a median of approximately 78.00 s) going through their first learning material with text and a figure, the 5th-grade pupils 115.38 s (with a median of approximately 81.00 s) and the 6th-grade pupils 152.54 s (with a median of approximately 122.50 s).

The total time taken to answer all test-lesson questions averaged about six minutes, which is equivalent to 355.55 s, with the 5th-grade pupils taking notably longer. The answering time for the 4th grade was 292.76 s, for the 5th grade 452.72 s, and for the 6th grade 275.90 s. There was a gender difference in answering times: 4th-grade girls used 156.16 s, while boys used 138.61 s; 5th-grade girls used 268.61 s, while boys used 246.76 s; and 6th-grade girls used 125.88 s, while boys used 155.16 s (Table 2).

**Table 2.** Average length of pupils' use of time to answer eight learning-material questions (in seconds).

|  | Mean Time Used for Answering | Standard Deviation | Standard Error | 1st Set Open Questions | 1st Set Multiple Choice | 2nd Set Open Questions | 2nd Set Multiple Choice | Pupils N |
|---|---|---|---|---|---|---|---|---|
| 4th-grade pupils | 292.76 | 131.78 | 18.64 | 133.38 | 29.92 | 90.60 | 38.86 | 50 |
| 5th-grade pupils | 452.72 | 202.25 | 35.75 | 198.56 | 82.38 | 110.69 | 61.09 | 32 |
| 6th-grade pupils | 275.90 | 126.38 | 23.88 | 103.68 | 41.00 | 90.29 | 40.93 | 28 |
| Average | 355.55 | 153.47 | 26.09 | 144.74 | 48.00 | 116.96 | 45.85 | 110 |

Regarding the answering of two open-ended questions about the learning material, the 4th-grade pupils took an average of 133.38 s (with a median of 107.50 s) to provide their responses, 5th-grade pupils took 198.56 s (with a median of 179.50 s), and 6th-grade pupils took 103.68 s (with a median of 66.50 s) (Table 2).

In contrast, for the two multiple-choice questions in the first part of the lesson, the 4th-grade pupils took an average of 29.92 s (with a median of 23.00 s) to answer, the 5th-grade pupils took 82.38 s (with a median of 64.50 s), and 6th-grade pupils took 41.00 s (with a median of 37.50 s).

As mentioned, in the second stage of the lesson, the pupils were reorganized to three groups according to the knowledge they had acquired about the topic studied in the first part. To do this, pupils' answers to questions in the test's first part were assessed by ChatGPT-3.5. If three to four answers out of four were correct (correctness rate was at least 75%), the pupil was placed into a higher knowledge group (i.e., from basic to intermediate, of from intermediate to advanced). If the answers were mostly incorrect (correctness rate was less than 50%), the pupil was downgraded to a lower knowledge group (i.e., from advanced to intermediate or from intermediate to basic). Otherwise, the pupils remained in the same group as they were in the beginning of the test (Figure 3). Then, for each group, a new text with a figure was provided, along with two open questions and two multiple-choice questions that were later evaluated.

As regards the second set of learning material, i.e., the text and one figure, on average, the 4th-grade pupils spent 91.56 s (with a median of 78.50 s) going through their learning material, the 5th-grade pupils took an average of 110.69 s (with a median of 106.50 s), and the 6th-grade pupils took an average of 98.82 s (with a median of 83.00 s). Regarding the two open-ended questions, the 4th-grade pupils took an average of 90.60 s (with a median of 74.50 s) to provide their responses, the 5th-grade pupils took an average of 181.50 s (with a median of 150.50 s), and the 6th-grade pupils took an average of 90.29 s (with a median of 84 s). In contrast, for the two multiple-choice questions, the 4th-grade pupils took an average of 38.86 s (with a median of 25.50 s) to answer, the 5th-grade pupils took an average of 61.09 s (with a median of 49.50 s), and the 6th-grade pupils took an average of 40.93 s (with a median of 32.00 s) (Table 2).

### 4.3. Conducting the Self-Evaluation Part of the Test Lesson

Upon completing the study-content part, the pupils were asked final questions regarding their experiences gained using the devices, i.e., the Ceibal laptop and Chromebook, and studying the required learning material. The first question gauged the ease of using the device to study the topic. Among the pupils, the responses were rather evenly divided, though they leant toward easiness of the use of the device to learn the topic, as only about one out of five pupils found it difficult to a certain extent: 29.1% found it very easy, 20.9% quite easy, 29.1% neither easy nor difficult, 12.7% quite difficult and 8.2% very difficult. At the 4th grade, these shares were 24.0%, 24.0%, 30.0%, 16.0% and 6.0%; at the 5th grade, 37.5%, 21.9%, 31.3%, 6.3% and 3.1%; and at the 6th grade, 28.6%, 14.3%, 25.0%, 14.3% and 17.9%.

Subsequently, the pupils provided their estimations regarding the difficulty level of the studied content, yielding a large variation among the pupils: 8.2% found it easy, 32.7% quite easy, 33.6% neither easy nor difficult, 20.0% quite difficult and 5.5% very difficult. At the 4th grade, the shares were 4.0%, 32.0%, 40.0%, 20.0% and 4.0%; at the 5th grade, 21.9%, 43.8%, 18.8%, 9.4% and 6.3%, and at the 6th grade, 0.0%, 21.4%, 39.3%, 32.1% and 7.1%. This suggests a need to properly implement the targeting of the learning material with generative AI to match the pupils' learning skills. In fact, there was a variation among pupils in different knowledge groups regarding the perception of difficulty of the learning material (Table 3).

**Table 3.** Pupils' perceived difficulty of the lesson contents in knowledge groups (%).

|  | Very Easy | Quite Easy | Neither Easy nor Difficult | Quite Difficult | Very Difficult |
|---|---|---|---|---|---|
| Basic knowledge group | 17.2 | 24.1 | 48.3 | 6.9 | 3.4 |
| Intermediate knowledge group | 2.8 | 37.5 | 27.8 | 25.0 | 6.9 |
| Advanced knowledge group | 22.2 | 22.2 | 33.3 | 22.2 | 0.0 |

Following that, the pupils were asked whether the figures included in the lesson material supported their learning. Among all the pupils, the majority found that the figures supported their learning: 26.4% answered that the figures were very helpful, 30.0% quite helpful, 27.3% were unsure, 10.0% not really helpful, and 6.4% not at all helpful. At the 4th grade, the shares were 18.0%, 34.0%, 38.0%, 6.0% and 4.0%, at the 5th grade, 46.9%, 28.1%, 9.4%, 12.5% and 3.1%, and at the 6th grade, 17.9%, 25.0%, 28.6%, 14.3% and 14.3%.

In the basic knowledge group, 37.9% found the images very helpful, 27.6% quite helpful, 24.1% unsure, 6.9% not really helpful and 3.4% not at all helpful. In the intermediate knowledge group, 19.4% found the images very helpful, 30.6% quite helpful, 30.6% unsure, 11.1% not really helpful and 8.3% not helpful at all. In the advanced knowledge group, 44.4% evaluated the images as very helpful, 33.3% quite helpful, 11.1% unsure, 11.1% not really helpful and 0.0% not helpful at all.

Subsequently, the pupils conducted a self-evaluation to determine the extent to which they perceived themselves to have learned about the topic covered in the lesson after finishing the test. The overall responses indicated that only one out of seven pupils mentioned that they had learned little or nothing, while the clear majority (57.2%) mentioned that they had learned much or very much. Among all pupils, 23.6% indicated that they had learned very much, 33.6% learned much, 29.1% learned somewhat, 9.1% learned little, and 4.5% learned not at all. There was a variation between the school classes, as pupils in the 5th grade perceived themselves to have learned more than those in the 4th and 6th grades. Overall, substantially more girls perceived that they had learned very much (33.0%) compared to boys (17.5%) (Table 4).

**Table 4.** Pupils' perception of learning during the lesson (%).

| | Very Much Learning | Much Learning | Some Learning | Little Learning | Not at All Learning | Pupils N |
|---|---|---|---|---|---|---|
| 4th-grade pupil | 20.0 | 40.0 | 28.0 | 10.0 | 2.0 | 50 |
| 5th-grade pupil | 34.4 | 31.3 | 28.1 | 3.1 | 3.1 | 32 |
| 6th-grade pupil | 17.9 | 25.0 | 32.1 | 14.3 | 10.7 | 28 |
| Boys | 17.5 | 33.3 | 38.1 | 7.9 | 3.2 | 63 |
| Girls | 33.3 | 35.6 | 17.8 | 8.9 | 4.4 | 45 |
| Much interest in the topic | 70.6 | 17.6 | 5.9 | 5.9 | 0.0 | 17 |
| Little interest in the topic | 23.5 | 47.1 | 29.4 | 0.0 | 0.0 | 17 |
| Not at all interested in the topic | 0.0 | 29.6 | 40.7 | 14.6 | 14.8 | 27 |
| Perceived having much or some knowledge on the topic | 42.2 | 36.4 | 15.2 | 3.0 | 3.0 | 33 |
| Perceived having little or no knowledge on the topic | 14.3 | 28.6 | 38.1 | 9.5 | 9.5 | 42 |
| Performed best in the test (over 75% answers correct) | 20.0 | 40.2 | 33.3 | 0.0 | 6.7 | 15 |
| Performed poorest in the test (less than 50% correct) | 24.1 | 24.1 | 34.5 | 10.3 | 6.9 | 29 |
| Enjoyed the test much or very much | 32.9 | 38.4 | 21.9 | 6.8 | 0.0 | 73 |
| Enjoyed the test little | 0.0 | 33.3 | 33.3 | 16.7 | 16.7 | 6 |
| Enjoyed the test not at all | 0.0 | 0.0 | 20.0 | 20.0 | 60.0 | 5 |
| Average | 23.6 | 33.6 | 29.1 | 9.1 | 4.5 | 110 |

Typical characteristics of those who perceived themselves to have learned a lot were pupils with excellent grades overall, who were initially placed into the advanced level in the beginning (determined by their Social Sciences rating) (t (DF = 16) = 28.625, $p$ = 0.027), and who showed interested in the subject (t (DF = 12) = 39.322, $p$ < 0.001). Those who perceived not to have learned at all were typically not interested about the subject, did not enjoy the lesson, and evaluated the content and exercises as "very difficult" or "difficult" (t (DF = 16) = 44.452, $p$ < 0.001). Among pupils who performed the best in the test (15 pupils who had a performance score of more than 80% correct answers), 60.0% perceived that they had learned very much or much, 33.0% perceived that they had learned somewhat, and 6.7% little or very little. Of those who performed weakest in the test (29 pupils having less than 50% correct answers), 48.2% perceived themselves to have learned very much or much, 34.5% to have learned somewhat and 17.2% little or very little. A positive correlation was found between pupils perceiving that they had learned much about the topic and their interest in the topic (r = 0.389, $p$ < 0.001) and enjoyment of the test lesson (r = 0.524, $p$ < 0.001) (Table 4). This suggests the vital importance of the design of the learning material and of generating interest among pupils toward the topic learned during a classroom session. Both are tasks that can be accomplished with generative AI.

The final feedback question asked the pupils about their level of enjoyment in learning the topic using the laptop and the generative AI-modified learning material (Table 5). The responses varied among the pupils: 40.0% mentioned they liked it very much, 26.4% much, 23.6% somewhat, 5.5% little, and 4.5% not at all. These percentages were, at the 4th, grade, 32.0%, 36.0%, 20.0%, 10.0%, and 2.0%; at the 5th grade, 59.4%, 15.6%, 21.9%, 0.0%, and 3.1%; and at the 6th grade, 32.1%, 21.4%, 32.1%, 3.6%, and 10.7%. Therefore, in general, more than two thirds enjoyed learning in the generative AI environment provided. As regards the test lesson, on average, girls enjoyed such learning slightly more than boys and enjoyment varied more among boys (Table 5). Also, earlier studies among primary and secondary school pupils found that in general, pupils were positive about the use of ChatGPT as a contemporary learning environment tool belonging to generative AI [14,15].

**Table 5.** Pupils' enjoyment of lesson with laptop and generative AI-modified learning material (1 = not at all ... 5 = very much).

| | Mean | Standard Deviation | Standard Error | Pupils N |
|---|---|---|---|---|
| 4th-grade boy pupils | 3.39 | 0.99 | 0.19 | 28 |
| 4th-grade girl pupils | 3.82 | 1.14 | 0.24 | 22 |
| 5th-grade boy pupils | 4.05 | 1.09 | 0.23 | 22 |
| 5th-grade girl pupils | 4.80 | 0.63 | 0.20 | 10 |
| 6th-grade boy pupils | 3.69 | 1.44 | 0.44 | 14 |
| 6th-grade girl pupils | 3.77 | 1.01 | 0.28 | 14 |

Pupils who liked the test lesson very much were typically very interested in the subject (t (DF = 16) = 34.198, $p$ = 0.005), found the readability of learning material to be very easy (t (DF = 16) = 39.127, $p$ = 0.001), and found the learning-material illustrations to be very helpful to support learning (t (DF = 16) = 81.844, $p < 0.001$). Those pupils that did not like the test lesson were typically not interested at all in the subject (t (DF = 16) = 34.198, $p$ = 0.005), did not learn at all (t (DF = 16) = 61.426, $p < 0.001$) and their learning material difficulty mean level was intermediate (t (DF = 8) = 16.727, $p$ = 0.033).

*4.4. Correctness of Pupils' Answers to Questions Regarding the Test-Lesson Learning Material*

The generative AI model was employed to determine the correctness of the pupils' answers, and each of the eight questions (four open-ended and four multiple-choice) regarding the learning material was evaluated as either correct or incorrect.

On average, the pupils could answer 4.66 out of the 8 learning material questions correctly, resulting in a share of 58.2% of correct answers and 41.8% incorrect ones. Pupils who perceived themselves as having little of no knowledge about the study topic, or those who perceived that they knew much about the topics, obtained noticeably fewer correct answers to questions than others. This suggests that proportionally higher share of pupils who perceived their knowledge to be at either extreme ends could not estimate their knowledge realistically (Tables 6 and 7).

**Table 6.** Pupils' correct answers to learning-material questions according to their perceived knowledge (%).

| | Correct Answers (N) | Correct Answers (%) | Pupils N |
|---|---|---|---|
| Perceived having much knowledge on the topic | 3.68 | 46.0 | 22 |
| Perceived having some knowledge on the topic | 4.65 | 58.1 | 20 |
| Perceived having fair knowledge on the topic | 5.14 | 64.3 | 35 |
| Perceived having little knowledge on the topic | 5.13 | 64.1 | 23 |
| Perceived having no knowledge at all on the topic | 4.10 | 51.3 | 10 |

No statistically significant correlation was observed between the pupils' higher grades in Social Sciences at school and their performance in answering the questions. Generally, those with the highest grades did not provide more correct answers compared to those with lower grades. The average correctness score for pupils with the highest grades in Social Sciences at school was 53% (with a median of 50%). Those with the lowest grades in Social Sciences obtained average scores of 56% (with a median of 60%). However, there was a positive correlation between the pupils' higher grades from all school subjects (r = 0.387, $p < 0.001$) and their self-perceived interest in the subject (r = 0.389, $p < 0.001$). In addition, there were statistically significant differences in Social Sciences grades in relation to pupils' self-evaluated learning ((t (DF = 16) = 28.625, $p$ = 0.027).

**Table 7.** Pupils' response time to learning-material questions and the correctness of answers.

| | Multiple-Choice Questions | | | Open-Ended Questions | | |
|---|---|---|---|---|---|---|
| | Correct Answers (%) | Correct Answers (N) | Total Answers (N) | Correct Answers (%) | Correct Answers (N) | Total Answers (N) |
| 11–20 s | 100.0 | 4 | 4 | - | - | - |
| 21–40 s | 92.6 | 63 | 68 | 25.0 | 1 | 4 |
| 41–80 s | 80.2 | 138 | 172 | 50.0 | 8 | 16 |
| 81–160 s | 72.2 | 93 | 128 | 36.5 | 35 | 96 |
| 161–320 s | 65.6 | 42 | 64 | 36.3 | 74 | 204 |
| 321–500 s | - | - | - | 41.3 | 33 | 80 |
| More than 500 s | 25.0 | 1 | 4 | 52.5 | 21 | 40 |

No statistically significant correlation was found between the pupils' expressed positive interest in the topic and their performance in answering the open and multiple-choice questions (r = 0.006, *p* = 0.954). On the contrary, pupils who expressed a high level of interest in the lesson topic generally achieved lower correctness scores compared to those who expressed only a little interest in the topic. For those who expressed the highest interest, the average correctness score was 3.9 answers correct out of 8 (with a median of 4.0/8). Those who expressed the lowest interest obtained scores of 4.5/8 (with a median 5.0/8). In addition, the percentage of correctness was found to be almost the same among girls (4.64/8) compared to boys (4.78/8). This suggests that having an interest in the topic may not necessarily lead to learning about the topic.

The answering times showed rather consistent patterns. In general, the shorter the total time used by the pupil to answer the multiple-choice question, the higher the share of correct answers. This suggests that pupils who knew the correct answer usually answered quickly. As regards open-ended responses, the situation was just the opposite: the shorter the total time used by the pupil to answer the open-ended questions, the lower the share of correct answers. This suggests that pupils who did not know the answer usually answered something short quickly, and that was often not a correct answer. Those who knew the answer elaborated on their responses, and these responses were usually correct (Table 7).

## 5. Discussion

There is an immense potential of generative AI in transforming school education, and the year of 2023 was only the very beginning. The findings of this article suggest that generative AI can be effectively used to differentiate pupils based on their prior knowledge and interest in the topic being studied in schools, and adjust the learning material accordingly both before and during the lesson.

In the test lesson, it was possible to divide pupils into basic, intermediate, or advanced knowledge groups. This adapted learning material was not limited to the text alone; it also included associated exercises and visual material that aided in learning. In addition, it was possible to receive immediate feedback of pupils' learning during the lesson and adjust the learning material in real time, taking into account the pupil's performance. This requires both further empirical analyses and theoretical reflection on learning processes. For example, it could draw from the flow theory to education during the generative AI era [39,43].

One key strength of utilizing generative AI lies in its ability to design learning materials that foster improved performance by leveraging pupils' motivation to learn. The concept of cognitive ergonomics plays a crucial role in supporting pupils' learning, irrespective of whether they initially belong to groups with higher or lower learning skills and motivation. This differentiation and the related adjustment of learning materials and tasks proved beneficial in promoting motivated learning among pupils and enhancing their performance. In addition, pupils' performance and their knowledge about the topic being learned are continuously monitored. As a result, teachers and education authorities are consistently

aware of their situation and can identify moments when specific assistance and intervention are needed. In the end, the necessity for exams, sometimes creating unnecessary pressures for learners and teachers, as knowledge-assessment mechanisms becomes less significant.

As of the summer of 2023, it is evident that further substantial development is required for generative AI to become a universally accessible and reliable tool that can be effectively utilized in classrooms worldwide. In addition, more studies need to be conducted with wider samples as well as through longitudinal material collection and in-depth statistical analyses across school grades, curricula and countries. These would make it possible to generalize the details regarding the reception of pupils to learning materials and how the materials adapted with generative AI tools impact pupils' learning results. Furthermore, teachers' and education managers' perspectives need to be taken into account for the smooth integration of generative AI into education. Despite the need for ongoing improvements, the rapid pace of development in generative AI makes it unwise to dismiss its use in education. In fact, generative AI in the realm of digital learning environments has the major potential to support broader sustainable development [20], including SDG4 [6]. It can support the provision of access to the newest relevant information for learning and in making that learning accessible and inclusive for all kinds of pupils with different knowledge levels and interests. This enables them to acquire learning skills and knowledge required for their current and future sustainable development. When properly developed and implemented, generative AI has the potential to revolutionize education by providing easy access to high-quality learning environments. This, in turn, can create impactful, inclusive, and sustainable learning in schools to benefit pupils, teachers and education managers alike, all of which is significant for sustainable development.

## 6. Conclusions

This article presents an examination of the application of generative AI in primary school education, specifically focusing on how pupils learn during a school lesson where the text, illustrations, and exercises have been designed and personalized using generative AI based on each pupil's knowledge and interest in the topic. This article, therefore, responds to the widely recognized need, expressed by scholars and education managers, to study how generative AI and ChatGPT function in school education practices, e.g., [7–9,16,32–34].

The study involved a total of 110 pupils, aged 8–14 years old, who utilized laptop computers to access the learning material. The research was conducted in Uruguay, specifically within the context of the Social Sciences (History) curriculum and a lesson for the 4th-, 5th- and 6th-grade pupils. The application used belongs the Digileac platform, which provides a digitally assisted learning environment supported by the use of generative AI [18].

This article provides a detailed explanation of the process involved in creating the lesson, learning material and the assessment methods employed. This information is crucial for understanding the procedures involved in designing future tests effectively. The overall number of analyzed pupils was rather limited and analysis was conducted only regarding one study topic, so for broader generalization, wider samples and other studies would be needed. Nevertheless, as indicated above, already this study indicates how generative AI can be easily and very effectively used to generate learning material for pupils with different knowledge bases on the topics of the lesson.

The analysis indicates that it was possible to use ChatGPT-3.5 and 4, as one example of generative AI, to personalize and customize learning material so that it would match the knowledge and learning skills of pupils with different levels of knowledge. Differences were observed among pupils regarding their reading time of the learning material as well as regarding their answering time to open-ended and multiple-choice questions. The time varied between genders and among pupils from the 4th to 6th grade; however, a more detailed analysis requires larger data. Overall, two out of three pupils enjoyed learning the generative AI modified material with the laptop, and only one out of ten liked it only a little or not at all. The clear majority of pupils mentioned having learned much or very much as the result of the test. In addition, there was a positive correlation with such

perceptions if the pupil was interested in the topic and perceived the learning experience positively. The latter suggests an important possibility for the use of generative AI in learning environments.

Based on the results of this study, generative AI, in this case ChatGPT-3.5 and 4, has the potential to effectively tailor the school learning-material content so that a diversity of pupils can be provided with personalized interactive learning materials and environments. This will enhance their learning engagement and result in motivating learning experiences that are cognitively ergonomic, encouraging them to move into deeper knowledge along BLOOM's taxonomy. This suggests that teachers need to play an important role in designing and implementing the ways generative AI is used so that all benefits can be taken advantage of.

As indicated in the Section 5, there is a need for longitudinal studies on the topic, as well as systematic pre-evaluation, evaluation, and post-evaluation regarding the use of generative AI in education. In particular, research-based evidence about generative AI's role and impacts in pupils' learning outcomes, including when learning material has been adapted with generative AI, is required.

**Author Contributions:** Conceptualization, J.S.J. and A.G.G.; methodology, J.S.J. and A.G.G.; software, J.S.J. and A.G.G.; validation, J.S.J. and A.G.G.; formal analysis, J.S.J. and A.G.G.; investigation, J.S.J. and A.G.G.; resources, J.S.J.; data curation, J.S.J. and A.G.G.; writing—original draft preparation, J.S.J. and A.G.G.; writing—review and editing, J.S.J. and A.G.G.; visualization, J.S.J. and A.G.G.; supervision, J.S.J.; project administration, J.S.J.; funding acquisition, J.S.J. All authors have read and agreed to the published version of the manuscript.

**Funding:** The research was partially funded by the University of Turku and by the Business Finland co-creation project Digileac nr. 4153/31/2022.

**Institutional Review Board Statement:** Not applicable.

**Informed Consent Statement:** Not applicable.

**Data Availability Statement:** Not applicable.

**Conflicts of Interest:** The authors declare no conflict of interest. The funders had no role in the design of the study; in the collection, analyses, or interpretation of data; in the writing of the manuscript, or in the decision to publish the results.

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
