# Peer review of "Generative AI and ChatGPT in School Children’s Education: Evidence from a School Lesson"

_sustainability, doi:10.3390/su151814025_

Round 1

Reviewer 1 Report

This article presents an investigation into the practical implementation of generative AI within primary school education. The primary focus of this study is centered on elucidating the dynamic process through which students engage with educational content during classroom lessons. Specifically, the article delves into the intriguing realm where text, illustrations, and exercises are meticulously crafted and tailored through the utilization of generative AI, specifically Chat GPT 3.5.

The research itself entailed the active participation of 110 pupils, ranging in age from 8 to 14 years old, all of whom made use of mobile devices and tablets as part of the pedagogical approach. The study was meticulously conducted within the educational context of Uruguay, with a specific focus on the History curriculum. Throughout the exploration, the article provides a detailed exposition of the intricate steps involved in curating these personalized lessons.

However, there lies an imperative within the results section to offer a more comprehensive and granular portrayal of the findings. For instance, in presenting the comparative outcomes of the administered tests, it is suggested that the following format be adhered to: t (DF=25) = 00.11, p < 0.000. This modification would ensure a standardized and precise representation of the statistical significance observed in the test results. Moreover, when conveying the outcomes of the correlation analyses, it is recommended that the ensuing format be followed: r(38) = .34, p = .032. This alteration would enhance the clarity and exactitude of the statistical correlations and their associated significance levels.

Referring the following publications could offer valuable insights and potential solutions in the context of your article: “Investigating the influence of an Arduino-based educational game on the understanding of genetics among secondary school students. https://doi.org/10.3390/su15086942” and “An experimental study on the effectiveness of a STEAM-based learning module in science education. https://doi.org/10.3390/su15086807.”

Author Response

REPLY TO THE REVIEWER

Thank you very much for your very constructive overall comments and the suggestion to make the statistical testing information more accurate, and the reference to two articles.

We revised the manuscript by by referring to the article regarding educational game. Other article did not discuss AI in education so that was slightly more distant regarding our research topic, and due to the length of our manuscript, we could not extend the discussion to that direction.

We also extended the statistical testing information. 

Reviewer 2 Report

The study has a timely topic. Researchers presented their rrsearch welll. Discussion can be improved with comparing and contrasting findings with relevant studies. In addition I suggest authors to add the screenshots of sample ai generated materials and tests. Practcal and theoretical implications should be explicitly addressed. 

Author Response

REPLY TO THE REVIEWER

Thank you very much for your very constructive and concise comments.

We considered the suggestion to add screenshots of materials and tests. However, these were conducted in Spanish so many readers could not receive proper information from these, so in the end we did not add these. However, our next tests are also in English-speaking environment, so this suggestion can be implemented then.

We revised discussion and conclusion to express practical and theoretical implications more clearly.

Comparing the results with other similar articles proved to be very difficult as we have not been able to identify any former research results based on tests on generative AI modified learning material for primary school pupils. The cases regarding small tests with university student on various topics are from very different context thus not providing real possibility for comparison.

Reviewer 3 Report

This article is devoted to one of the topics relevant for sustainable development, namely, the use of artificial intelligence in teaching. The research methodology corresponds to the set goal. In general, the involvement of sources covering this issue allows us to identify the main problems of the study. However, the list of references can be expanded. The topic of artificial intelligence and its application in education refers to a wider range of topics. It is necessary to consider in more detail the issue of teacher competencies. Pros, cons and challenges of using artificial intelligence. These are topics that overlap with this article. It is somewhat unclear how the research questions presented on lines 109-110 (page 3) are solved. In conclusion, it is necessary to tell the reader about this. The conclusion in the Abstract does not correspond to the Conclusion in the text. The last sentence of the Abstract is obvious without research. The authors need to show what they have come to. And they have something to show. This article is the result of an interesting research relevant to the current state of education and its development.

Author Response

REPLY TO THE REVIEWER

Thank you very much for your very clear and constructive comments.

We added few comments more regarding the suggestion to discuss about teachers competences in the generative AI context. We agree that the list of general references to generative AI in education could be expanded. However, to keep the article within the general length, we selected the most significant overall references to that topic. In addition, in the manuscript we do not discuss at length the overall use of generative AI in education but how generative AI can be used to adapt the learning material for pupils.

In sub-chapter 2.2. Current practices of generative AI in education in schools, we present a concise discussion of challenges and opportunities with this regard and refer to comprehensive overviews of the challenges and risks associated with the use of LLMs in education [see Kasneci et al., 2023].

Following the reviewer's suggestion, we revised also the conclusions so that it contains more clearly our finding regarding the research questions.

Abstract and conclusions are now more clearly connected to each other. We agree that the last sentence of the abstract is general (generative AI tools need to be developed, refined and optimized), however, that is also a result from out empirical study regarding the use of ChatGPT to adapt pupils’ learning material.

Reviewer 4 Report

1. The title subject is extremely broad. The title does not indicate the pertinent objectives and questions of the research.

2. The primary concerns of this study were as follows: Not applicable; informed consent statement of the institutional review board. How can this research be conducted on children under the age of 18? How is this research valid? "The primary data source for this study is the responses of 110 students aged 8 to 14 who participated in the test lesson as part of their ongoing annual curriculum in a school classroom. The 4th grade respondents (50 students) were 8–10 years old, the 5th grade respondents (32) were 11 years old, and the 6th grade respondents (28) were 11–14 years old. This investigation required an IRB and parental consent forms.

3. The objectives of the study were unclear. The research questions were not fundamental contributions to any field of sustainability. How does this study demonstrate the long-term viability of education or theoretical learning?

4. Why and how were the evaluation questions developed for Section 4? There was no evidence to suggest that all questions were required for this research.

5. All of the results were merely basic statistics. Numerous techniques, including SEM, PLS-SEM, ANOVA, T-TEST, and others, are available for demonstrating significant factors and concerns. However, the authors mention a few in Section 4.2 but provide no literature reviews.

6. The conclusion was unrelated to the research questions.

7. The most significant issue in this study was the lack of merit or contribution from sustainable studying, learning, or the use of AI to support education. There are currently issues with AI in education. How do you verify that the material or information provided by AI is true and accurate? It does not matter how the AI generates class material; what the students do correctly learn is more important.

8. The conclusion is not entirely the opinion of the author. The conclusion of the study does not provide conclusive evidence.

There were some moderately incorrect English sentences, particularly in section 4.2, as well as numerous errors in punctuation, articles, and complex sentences.

Author Response

REPLY TO THE REVIEWER

Thank you very much for your very clear comments and numerous suggestions how to improve the manuscript. We have reflect each of them and revised the manuscript to the extent possible.

  1. The title subject is extremely broad. The title does not indicate the pertinent objectives and questions of the research.

We agree that the title is broad. However, as the manuscript indicates, in the introduction and conceptual part and discussion and conclusions we discussed generative AI and education from such a broad perspective. We agree that the empirical study regards only a smaller part of the use of generative AI in education. However, finally we considered that the broader title gives a better description of the article and the article is found easier through search engines rather than a very specific title referring only to the empirical study.   

  1. The primary concerns of this study were as follows: Not applicable; informed consent statement of the institutional review board. How can this research be conducted on children under the age of 18? How is this research valid? "The primary data source for this study is the responses of 110 students aged 8 to 14 who participated in the test lesson as part of their ongoing annual curriculum in a school classroom. The 4th grade respondents (50 students) were 8–10 years old, the 5th grade respondents (32) were 11 years old, and the 6th grade respondents (28) were 11–14 years old. This investigation required an IRB and parental consent forms.

Manuscript is now slightly revised to take into account this observation. As explained in the manuscript, prior to conducting the test, the directors of the participating schools granted permission, and the teachers were duly informed about the testing process, including the research integrity and ethical guidelines. To guarantee that all these details and procedures were meticulously implemented, one of the authors personally observed the test procedures on-site, while all test materials were automatically collected and recorded through the designated test platform. To ensure anonymity, the pupils answered the test without providing their names. The pupils took part of the test voluntarily, they had the right the withdraw from the test at any moment or to leave questions without answers. In fact, the material contains only results from those pupils who attended the whole test lesson and answered to the questions. The manuscript is part of broader research regarding generative AI in education and related procedures with research integrity and ethical board is implemented. 

  1. The objectives of the study were unclear. The research questions were not fundamental contributions to any field of sustainability. How does this study demonstrate the long-term viability of education or theoretical learning?

Following the reviewer’s suggestion, we revised the manuscript to connect the topic even more to sustainability, namely SDG4.

The reviewer’s suggestion to reflect on long-term viability of education and theoretical learning is very relevant and for this we revised discussion and conclusion.

To our understanding, this study is globally among the first if not the first in which have been used generative AI to design the primary school material and conducting an analysis of its viability, therefore these study objective have been expressed rather broadly. We consider that generative AI is very relevant to reach SDG4 (education) that we also elaborate in the manuscript. Obviously, as open source generative AI was launched only in November 2022, that topic was not mentioned in SDGs that were designed in the mid-2010s. 

The basic objective of the study has been expressed in 96-101 (the utilization of generative AI in school education, aiming to contribute to the understanding and exploration of its potential in educational settings ... examines the design of a school lesson using generative AI, encompassing text, figures, and exercises, as well as the evaluation of pupils' learning through its implementation).

In 108-114 we express the more precise reserch questions regarding the empirical study (The research questions were: how generative AI was utilized to classify learning material and pupils into basic, medium, and advanced knowledge groups? What was the amount of time pupils with diverse backgrounds spent on learning the provided topics? How and how much pupils learned as evaluated by generative AI? What were the perceptions of pupils regarding the learning process with generative AI material and the devices used?)

  1. Why and how were the evaluation questions developed for Section 4? There was no evidence to suggest that all questions were required for this research.

Line 409-428 explains how the evaluation this questions were developed: “Once the original text had been validated in both English and Spanish, additional translations were performed using ChatGPT-3.5 and ChatGPT-4 to assess the results. Both of these generative AI models demonstrated the ability to accurately translate the content by employing the same prompt, yielding outputs that closely resembled human translation. To minimize randomness in the output text, the temperature was limited, which proved crucial in avoiding potential hallucinations of text content in zero-shot to few-shot scenarios. Notably, there was a key difference in the format of the answers produced by developed NLP models. ChatGPT-3.5 exhibited the freedom to choose the placement of the author and link source, while ChatGPT-4 displayed consistency in adhering to the original text format provided.“

We agree that all questions asked from pupils would not have been necessary for this article. In fact, they are use for another article about this topic.

  1. All of the results were merely basic statistics. Numerous techniques, including SEM, PLS-SEM, ANOVA, T-TEST, and others, are available for demonstrating significant factors and concerns. However, the authors mention a few in Section 4.2 but provide no literature reviews.

We agree with reviewer’s observation. An alternative could have been to elaborate the analysis more with testing of statistical significance of certain findings, as we did in some sections that the reviewer also noticed. However, the data (110 pupils) is rather small to elaborate advanced statistical testing, especially when the data would be divided into sub-units. However, this suggestion is very valuable and we will definitely implement it in the next articles that will be based on larger data that we are currently collecting.

  1. The conclusion was unrelated to the research questions.

This reviewer’s observation is very relevant, thus we revised conclusion so that it contains explicit responses to all research questions that were mentioned in the introduction.

  1. The most significant issue in this study was the lack of merit or contribution from sustainable studying, learning, or the use of AI to support education. There are currently issues with AI in education. How do you verify that the material or information provided by AI is true and accurate? It does not matter how the AI generates class material; what the students do correctly learn is more important.

The reviewer makes here very important observations. However, as regards the accuracy and truthfulness of the study material adapted by generative, this is already explained in the manuscript – namely that the master text was designed from reliable and verified sources and that the potential hallucinations (mistakes) in the text were eliminated by us the authors.

We also verify challenges and possibilities of using generative AI for adapting learning material and evaluating pupils’ performance that is a novel and major contribution of generative AI to education. To our understanding, this manuscript is globally among the first if not the first study that have been able to study this in empirical context of the primary school.

Obviously, as the reviewer suggests, there are issues with AI in education, and here we focus on generative AI, as discussed in the conceptual part.

We also revised the manuscript to indicate more clearly the contribution of generative AI to sustainability, namely SDG4 (education).

  1. The conclusion is not entirely the opinion of the author. The conclusion of the study does not provide conclusive evidence.

We have now revised conclusion so that it responses more directly to our research questions. We acknowledge that the findings are not conclusive, thus we emphasize (in discussion) the need for a longitudinal study to establish more definitive results. However, the results derive from a generative AI education context that has barely studied so far.

There were some moderately incorrect English sentences, particularly in section 4.2, as well as numerous errors in punctuation, articles, and complex sentences.

We have revised the text language-wise to the extent possible, particularly section 4.2, and we apologize for not being at the sophisticated level of native English speakers.  

Round 2

Reviewer 4 Report

1. The authors confirmed consistently using the title. The title was overly inclusive and did not apply to all information, locations, religions, nations, languages, and behaviors. Therefore, I entirely disagree with the authors that the title was appropriate and supported by sound rationale. In addition, the samples did not correspond with the title. Assume the population was 600,000,000, with 95% confidence, 5% error, and n=385. This study of 110 students was inaccurate, valid, and reliable. The methodology was, therefore, flawed and inaccurate.

2. Regardless of the study concerns or the research conducted, the consent form was required, demonstrating that the study adhered to research ethics. The provided consent form contained false and inaccurate information. Therefore, this study was not suitable for publication as a journal article. Except for the curriculum study, no one may collect data directly from children and use it for public research without parental consent. 

Please see :
https://www.wma.net/policies-post/wma-declaration-of-helsinki-ethical-principles-for-medical-research-involving-human-subjects/

https://researcher.life/blog/article/ethical-declarations-for-journal-submission/#:~:text=Submit%20ethical%20declarations%20that%20acknowledge,be%20construed%20as%20conflicts%20of

For the consent form, please see :
https://research.unc.edu/human-research-ethics/consent-forms/

3.  Authors claimed “The basic objective of the study has been expressed in 96-101 (the utilization of generative AI in school education, aiming to contribute to the understanding and exploration of its potential in educational settings ... examines the design of a school lesson using generative AI, encompassing text, figures, and exercises, as well as the evaluation of pupils' learning through its implementation).

In 108-114 we express the more precise reserch questions regarding the empirical study (The research questions were: how generative AI was utilized to classify learning material and pupils into basic, medium, and advanced knowledge groups? What was the amount of time pupils with diverse backgrounds spent on learning the provided topics? How and how much pupils learned as evaluated by generative AI? What were the perceptions of pupils regarding the learning process with generative AI material and the devices used?)”

What lessons, how long, what material orientation, how many pages, what font size, what languages, what effective design, what comprehension, what possibilities, how many figures, and more... These were objectives-related issues. Therefore, it must be precisely defined and highly focused for this study. 

The second and third research questions were utterly unrelated to the topic.

4. Why and how were evaluation questions developed for significance? Using these questions, authors should provide either baseline research or relevant research. I am unaware of any evidence. T-test requires a minimum sample size of 10 individuals, so this is the smallest sample size allowed. 

5. There was still no formal methodology employed in this investigation. This study drew the majority of its conclusions from fundamental statistical information. Typically, basic statistics were used for descriptive and demographic purposes.

There were some minor mistakes in English sentence structure and numerous errors in punctuation, articles, and ambiguous "it" sentences.

Author Response

RESPONSE: Thank you for clarifying your viewpoints. We have now language edited the entire manuscript as you suggested, modified the article title as suggested, clarified the research integrity issues as suggested (following the national and university guidelines of which the documentation has been sent to the publisher) and revised the version to the extent possible, including being more precise with the research questions. We did not understand all your comments and some might have derived from the initial submission and not from the revised version.

  1. The authors confirmed consistently using the title. The title was overly inclusive and did not apply to all information, locations, religions, nations, languages, and behaviors. Therefore, I entirely disagree with the authors that the title was appropriate and supported by sound rationale. In addition, the samples did not correspond with the title. Assume the population was 600,000,000, with 95% confidence, 5% error, and n=385. This study of 110 students was inaccurate, valid, and reliable. The methodology was, therefore, flawed and inaccurate.

RESPONSE: Thank you for your comment clarifying your viewpoint. We have now added a subtitle “Evidence from a school lesson with ChatGPT use“. It specifies the detailed content of the empirical case. However, as we discuss generally the role of generative AI in education, we suggest to keep this broader theme in the title.

  1. Regardless of the study concerns or the research conducted, the consent form was required, demonstrating that the study adhered to research ethics. The provided consent form contained false and inaccurate information. Therefore, this study was not suitable for publication as a journal article. Except for the curriculum study, no one may collect data directly from children and use it for public research without parental consent. 

Please see :
https://www.wma.net/policies-post/wma-declaration-of-helsinki-ethical-principles-for-medical-research-involving-human-subjects/
https://researcher.life/blog/article/ethical-declarations-for-journal-submission/#:~:text=Submit%20ethical%20declarations%20that%20acknowledge,be%20construed%20as%20conflicts%20of For the consent form, please see :
https://research.unc.edu/human-research-ethics/consent-forms/

RESPONSE: Thank you for your comment clarifying your viewpoint. As regards the research integrity and ethics of the study conducted, we follow strictly and fully both the national research integrity guidelines and ethical committee guidelines as well as those of our university. The letter signed by the ethical review board has been forwarded to the publisher indicating all aspects of this. However, to clarity this, we have now added more detailed information regarding the research integrity and ethics of the empirical study. This is now highlighted in the text with yellow.

  1. Authors claimed “The basic objective of the study has been expressed in 96-101 (the utilization of generative AI in school education, aiming to contribute to the understanding and exploration of its potential in educational settings ... examines the design of a school lesson using generative AI, encompassing text, figures, and exercises, as well as the evaluation of pupils' learning through its implementation).

In 108-114 we express the more precise research questions regarding the empirical study (The research questions were: how generative AI was utilized to classify learning material and pupils into basic, medium, and advanced knowledge groups? What was the amount of time pupils with diverse backgrounds spent on learning the provided topics? How and how much pupils learned as evaluated by generative AI? What were the perceptions of pupils regarding the learning process with generative AI material and the devices used?)”

What lessons, how long, what material orientation, how many pages, what font size, what languages, what effective design, what comprehension, what possibilities, how many figures, and more... These were objectives-related issues. Therefore, it must be precisely defined and highly focused for this study. 

The second and third research questions were utterly unrelated to the topic.

RESPONSE: Thank you for your comment. In the manuscript, we already explained in detail how the empirical study was conducted: what was the original test lesson learning material, what language was used, which study subjects from 4th to 6th class the learning material topics belonged to, what were the topics, what were the devices used, how generative AI was used for modifying the learning material (text, figures, questions), how many pupils completed the test lesson, how much time pupils used to read the learning material and respond to the questions, etc. The topic and language and many other aspects are also mentioned in the article abstract and keywords. Obviously, one could describe even more of very many details but we do not think the readers would be interested in such very precise details in this much more general and broader article. Furthermore, we discuss about the process of generative AI mechanism and related technical aspects in the learning material transformation in another submitted manuscript to which we refer to in this manuscript as well.

As regards the second and third research questions, we revised the second and third research questions (Did the time length spent on learning generative AI modified material differ among pupils with diverse backgrounds? What perceptions pupils had on learning with generative AI modified material and the devices used?). We consider these important at this globally very early stage of the research about generative AI in schools to inform the readers, for example, that even if generative AI provided a slightly more complex text for pupils in more advanced knowledge level groups, the reading and answering times did not differ much from other knowledge level groups. It is also fundamental to study precisely the reading and answering times as it indicates if and how many pupils answer quickly because it is their style, etc. Also, it is relevant to study which ways pupils perceived the use of the generative AI adapted learning material to be able to fine tune next test lessons in more detailed way and suggest ideas for other scholars.

As mentioned, this is among the first tests globally in which generative AI modifying learning material has been tested in a school environment and it requires an enormous effort even to create such technological setting, so everything cannot be put in this first test, and the length of this manuscript is already well over 12,000 words. Again, as mentioned earlier, many details not addressed here are discussed in our manuscripts submitted to other journals.

  1. Why and how were evaluation questions developed for significance? Using these questions, authors should provide either baseline research or relevant research. I am unaware of any evidence. T-test requires a minimum sample size of 10 individuals, so this is the smallest sample size allowed. 

RESPONSE: Thank you for your comment. Unfortunately, we do not precisely understand the early part of this comment regarding evaluation questions, baseline research, relevant research, relevance and evidence. If you refer to what was asked from pupils, this is explained in the manuscript: namely relevant questions about participants’ demographic and scholarly backgrounds, their user perceptions, etc. Evidence comes from analysis results and differences observed among respondents. Besides descriptive statistics, we also conducted correlation analyses to the extent possible as indicated in the manuscript. The studied sample is rather small, therefore we could not go in to detailed statistical testing of sub-groups such as studying differences in gender per performance per knowledge group level etc. as then the sample size would have become too small (less than 5 persons within cells) as you rightly point out.

  1. There was still no formal methodology employed in this investigation. This study drew the majority of its conclusions from fundamental statistical information. Typically, basic statistics were used for descriptive and demographic purposes.

RESPONSE: Thank you for your comment. Unfortunately, we do not understand what here is meant by claiming that there was not a formal methodology. The research here is based on research questions that derive from learning theories and early ideas expressed in articles cited about generative AI in education. The research questions were answered through the quantitative analysis of collected empirical material. If by not having a formal methodology you mean that we did not postulate hypotheses, then yes, we agree. However, our intention was never to apply a hypothetic-deductive methodology that we did not consider suitable for this theory, methods and analysis.

Agree, in conclusions, we respond to research questions based on the analytical research results. The journal suggests a clear division of what should be written to discussion and what to conclusion, and such division of labor was also suggested by two other reviewers and we followed these suggestions. Conceptual reflections and broader issues are developed more in the discussion section.

Comments on the Quality of English Language

There were some minor mistakes in English sentence structure and numerous errors in punctuation, articles, and ambiguous "it" sentences.

RESPONSE: Thank you for your comment. We went through again the manuscript. Most likely we were able to pay attention to articles, punctuations and referred “it” sentences and to correct the manuscript, and enhance thus the overall legibility of the text. Though we used native speaker correction, not even all native speakers agree on the necessity of all punctuations and some indeterminate or indeterminate articles.

Round 3

Reviewer 4 Report

The authors have made numerous revisions to the manuscript. However, the manuscript was filled with severe aspectual problems.

1. The title of the manuscript is vague and goes beyond the research findings. In the context of "Generative AI and ChatGPT in the education of schoolchildren," "Generative AI" refers to the algorithms (such as ChatGPT) used to generate new content. Thus, what the title intends to convey. In addition, the words for "a school lesson" were ambiguous. Are they learning the school environment? Obtaining a broad education in school? What are the specific objectives?

2. The abstract states, "The results demonstrated that it was possible to use ChatGPT-3.5, as an example of generative AI, to personalize learning material so that it would meet the knowledge and 17 skills of pupils with varying levels of knowledge." However, I have not seen any reference to 17 skills in this study. 

3 The n sample of 110 students can only represent the population of approximately 175 students with a 95% level of confidence and a 5% margin of error, according to the abstract: "The study included 110 students aged 8 to 14 who were enrolled in grades 4 to 6 in four classes at two schools." The conclusion for the entire population in the topic of focus is unacceptable. The methodology is, therefore, inaccurate, invalid, unreliable, and inappropriate for the title. The n sample can only represent the particular location and problems, focusing on the group and conclusion. Based on the study's focus, methodology, and title, I am afraid I have to disagree with the authors.

4. This study has not demonstrated all levels of motivated learning and skill development, as stated in the abstract: "There is a promising potential for the use of generative AI in school education to support pupils' learning motivation and skill development." There is only one word for skill development in the manuscript. What and how does the manuscript's motivated learning and skill development contribute? 

5. In lines 108-110, "This case study examines the design of a school lesson using generative AI, including text, figures, and exercises, as well as the evaluation of pupils' learning through its implementation," there were no examples of the design of text, figures, and exercises as well as the evaluation of a generative AI-based lesson. No design explanation was provided in the manuscript. How is the design aligned with the SGD4 guidelines? a. How is the design compliant with SGD4? How does the evaluation of the design to ensure safety comply with SGD4? How the design evaluation is conducted.

6. The authors have not modified the manuscript and responded to the questions: How to verify the accuracy of all information generated by Generative AI? How do authors ensure that lessons for children are free of inaccuracy, inappropriateness, and lack of knowledge for appropriate reading levels?

7. Still issues, the authors persistently confirm that this study does not need IRB in Finland. Therefore, the study's consent form must be distributed before conducting the research. I have not seen a consent form for this study's practice. The consent form submitted was inaccurate and insufficient for this study. Thus, the research is indeed flawed.

8. The research methodology is inadequate in terms of evidence and relative conclusions for all students in grades 4-6. Only basic statistics were used to conclude the results. There were no demonstrated factors of studying, what multiple choices were used, how students interacted with the computer interface, what details of subjects and lessons, how often the tests were taken, what times were set, where and when parents were notified of this study….

9. There were no hypotheses for testing provided. 

10. The authors' discussions were primarily grounded in speculative rather than empirical evidence, leading to inconclusive findings.  

11. Many articles were presented at conferences with merits and intriguing concerns regarding generative AI in education. This study appears to generate content using a generative AI (ChatGPT) and distribute it to students in grades 4-6 without parental consent, proofreading, verification, or lesson-related connections. Then, a simple statistic was used to demonstrate the mean, standard deviation, standard error, maximum, and minimum conclusion. The manuscript contained fewer aspectual contributions (theoretical, implication, framework, suggestions for use, guideline, and future research based on this study) than typical journal articles of high-quality accuracy.

There were numerous punctuation, article, parenthesis, and grammar errors, as well as numerous sentences with ambiguous meanings.

Author Response

WE HAVE NOW ADDRESSED EACH COMMENT OF THE REVIEWER AND RESPONDED TO THEM, AND REVISED THE MANUSCRIPT ACCORDINGLY.

The authors have made numerous revisions to the manuscript. However, the manuscript was filled with severe aspectual problems.

  1. The title of the manuscript is vague and goes beyond the research findings. In the context of "Generative AI and ChatGPT in the education of schoolchildren," "Generative AI" refers to the algorithms (such as ChatGPT) used to generate new content. Thus, what the title intends to convey. In addition, the words for "a school lesson" were ambiguous. Are they learning the school environment? Obtaining a broad education in school? What are the specific objectives?

ANSWERS: THANK YOU FOR YOUR COMMENT. IN THE MANUSCRIPT, FROM THE BEGINNING TO THE END, ARE DISCUSSED GENERATIVE AI AND ITS USE IN EDUCATION, AND IN SCHOOLS, AND IN PARTICULAR REGARDING SCHOOL CHILDREN’S LEARNING. IN ADDITION, THE EMPIRICAL CASE REGARDS A TEST LESSON CONDUCTED IN A SCHOOL. WE FIND THE TITLE ACCURATE AND INFORMATIVE.

  1. The abstract states, "The results demonstrated that it was possible to use ChatGPT-3.5, as an example of generative AI, to personalize learning material so that it would meet the knowledge and 17 skills of pupils with varying levels of knowledge." However, I have not seen any reference to 17 skills in this study. 

ANSWERS: THANK YOU FOR YOUR COMMENT. THE NUMBER 17 REFERS TO NUMBERING OF LINE (17) THAT HAS BEEN GENERATED BY THE MACHINE. WE DO NOT CLAIM ANYTHING ABOUT 17 SKILLS.

3 The n sample of 110 students can only represent the population of approximately 175 students with a 95% level of confidence and a 5% margin of error, according to the abstract: "The study included 110 students aged 8 to 14 who were enrolled in grades 4 to 6 in four classes at two schools." The conclusion for the entire population in the topic of focus is unacceptable. The methodology is, therefore, inaccurate, invalid, unreliable, and inappropriate for the title. The n sample can only represent the particular location and problems, focusing on the group and conclusion. Based on the study's focus, methodology, and title, I am afraid I have to disagree with the authors.

ANSWERS: THANK YOU FOR YOUR COMMENT. AS INDICATED CLEARLY IN THE MANUSCRIPT, THE DATA IS CONSISTED OF 110 PUPILS FROM 4TH-6TH GRADES, AND WHO FOLLOWED THE LESSON AND ANSWERED TO THE QUESTIONS. THEREFORE, THE SAMPLE REFERS TO ALL (100% POPULATION) THESE PUPILS AND THE EMPIRICAL ANALYSIS IS ONLY ABOUT THESE PUPILS. WE DO NOT ANYWHERE CLAIM THAT THE EMPIRICAL RESULTS CAN BE DIRECTLY EXTRAPOLATED TO THE ENTIRE WORLD POPULATION AND THAT HAS NOT BEEN THE AIM IN THE ARTICLE AT ALL. THE TITLE, ABSTRACT, MATERIAL, AND RESULTS INDICATE VERY CLEARLY THIS AND FROM WHERE THE RESULTS COME FROM.

  1. This study has not demonstrated all levels of motivated learning and skill development, as stated in the abstract: "There is a promising potential for the use of generative AI in school education to support pupils' learning motivation and skill development." There is only one word for skill development in the manuscript. What and how does the manuscript's motivated learning and skill development contribute? 

ANSWERS: THANK YOU FOR YOUR COMMENT. IT IS IMPOSSIBLE TO STUDY ALL LEVELS OF MOTIVATED LEARNING IN ONE ARTICLE. IT HAS NOT BEEN OUR INTENTION, AND WE HAVE NEVER CLAIMED ANYTHING ABOUT ALL LEVELS OF MOTIVATED LEARNING. HOWEVER, IT IS POSSIBLE CONDUCT RESEARCH ABOUT MOTIVATED LEARNING DESPITE NOT COVERING ALL POSSIBLE ASPECTS OF IT, AS THIS WE HAVE DONE. AS REGARDS THE SKILLS DEVELOPMENT, THEY ARE DEFINED IN THE ARTICLE AS PUPILS LEARNING SKILLS (ALSO MENTIONING WITH PRECISION WHAT ARE INCLUDED IN THESE). THE MANUSCRIPT IS NOW MODIFIED BY ADDING THE WORD “LEARNING” TO “SKILLS” SO THAT IS MORE CLEARLY UNDERSTANDABLE.

  1. In lines 108-110, "This case study examines the design of a school lesson using generative AI, including text, figures, and exercises, as well as the evaluation of pupils' learning through its implementation," there were no examples of the design of text, figures, and exercises as well as the evaluation of a generative AI-based lesson. No design explanation was provided in the manuscript. How is the design aligned with the SGD4 guidelines? a. How is the design compliant with SGD4? How does the evaluation of the design to ensure safety comply with SGD4? How the design evaluation is conducted.

ANSWERS: THANK YOU FOR YOUR COMMENT. FOR EXAMPLE, RAWS 59-65 AND 820-829 EXPLICITY ILLUSTRATE THE EXAMPLES HOW GENERATIVE AI COULD CONTRIBUTE TO SDG4. THE DESIGN OF THE TEST LESSON TEXT IS IN DETAIL EXPLAINED IN 439-455 AND 572-579. THE OVERALL AIM OF USING GENERATIVE AI IN EDUCATION WOULD BE TO SUPPORT THE ACCOMPLISHMENT OF SDG4 AS IT IS EVIDENT FROM THE MANUSCRIPT.

IN CASE THE READER DOES NOT KNOW WHAT MEAN THE CONCEPTS “TEMPERATURE” OR “HALLUCINATION”, THESE ARE NOW EXPLAINED IN DETAIL IN THE TEXT.

DIRECT EXAMPLES OF THE LESSON TEXT, QUESTIONS OR ANSWERW ARE NOT ILLUSTRATED HERE AS THEY WERE IN SPANISH AND THAT WOULD NOT HAVE HELPED MOST READERS TO UNDERSTAND THEM. HOWEVER, NOW IT IS MORE CLEARLY MENTIONED WHAT IS AN OPEN-ENDED QUESTION AND WHAT IS A MULTIPLE-CHOICE QUESTION THOUGH THIS IS USUALLY UNDERSTOOD AMONG SCHOLARS IN SCIENTIFIC JOURNALS SUCH AS SUSTAINABILITY.

  1. The authors have not modified the manuscript and responded to the questions: How to verify the accuracy of all information generated by Generative AI? How do authors ensure that lessons for children are free of inaccuracy, inappropriateness, and lack of knowledge for appropriate reading levels?

ANSWERS: THANK YOU FOR YOUR COMMENT. ALREADY EARLIER WAS IN DETAIL EXPLAINED THE SELECTION AND THE DESIGN OF THE TEST LESSON TEXT, IN DETAIL IN 425-455 AND 572-579. THIS ALSO INDICATES VERY CLEARLY HOW THE PROCESS WENT THROUGH, INLUDING THE VERIFICATION OF THE MATERIAL ACCURACY. THE EMPIRICAL PROCESS THEN EXPLORES VERY DETAILLY, HOW THE SHARE OF CORRECT / INCORRECT ANSWERS WERE USED AS A BASE TO POTENTIALLY RELOCATE PUPILS INTO DIFFERENT KNOWLEDGE LEVEL GROUPS. WE HAVE NEVER CLAIMED IN THE ARTICLE ABOUT GENERATIVE AI’S COMPETENCES TO PROVIDE ALL CORRECT MATERIAL EVERYWHERE IN THE WORLD.

  1. Still issues, the authors persistently confirm that this study does not need IRB in Finland. Therefore, the study's consent form must be distributed before conducting the research. I have not seen a consent form for this study's practice. The consent form submitted was inaccurate and insufficient for this study. Thus, the research is indeed flawed.

ANSWERS: THANK YOU FOR YOUR COMMENT. AS ALREADY EARLIER MENTIONED AND CLEARLY INDICATED IN THE MANUSCRIPT AS WELL AS IN A SEPARATE CERTIFICATE BY THE ETHICS BOARD SENT TO THE PUBLISHER UNDERLINE AND OUR RELATED EXPLANATION TO THE PUBLISHED EXPLAINS, WE HAVE FOLLOWED ALL NATIONAL AND LOCAL RESEARCH INTEGRITY AND ETHICS GUIDELINES TO CONDUCT THIS RESEARCH, INCLUDING CONSENT-RELATED ASPECTS. 

  1. The research methodology is inadequate in terms of evidence and relative conclusions for all students in grades 4-6. Only basic statistics were used to conclude the results. There were no demonstrated factors of studying, what multiple choices were used, how students interacted with the computer interface, what details of subjects and lessons, how often the tests were taken, what times were set, where and when parents were notified of this study….

ANSWERS: THANK YOU FOR YOUR COMMENT. AS WE HAVE ALREADY MENTIONED, IN THE MANUSCRIPT IS INDICATED WITH UTMOST PRECISION HOW THE TEST WAS CONDUCTED, WHAT MATERIAL AND DEVICES WERE USED, WHAT WERE THE LESSON TOPICS IN EACH CLASS, HOW MANY TESTS WERE TAKEN, HOW LONG EACH TEST TOOK PLACE, WHAT WERE THE PUPILS PERCEPTIONS OF THE MATERIAL AND DEVICES USED, ETC.

YES, WE HAVE USED CORRECTLY BASIC STATISTICAL ANALYSES AND METHODS, INDLUDING IN MANY OCCASIONS THE TESTING OF THE RESULTS’ STATISTICAL SIGNIFICANCE WHEN THAT WAS RELEVANT AND USEFUL FOR THE OBSERVATIONS MADE.

  1. There were no hypotheses for testing provided. 

ANSWERS: THANK YOU FOR YOUR COMMENT. AS WE ALREADY MENTIONED, THE HYPOTHETIC-DEDUCTIVE APPROACH IS ONLY ONE POSSIBLE METHODOLOGICAL RESEARCH FRAMEWORK. THIS STUDY HAS NEVER BEEN FRAMED ALONG HYPOTHETIC-DEDUCTIVE APPROACH. THEREFORE, WE DO NOT STATE HYPOTHESIS THAT SHOULD BE VERIFIED OR REJECTED.

  1. The authors' discussions were primarily grounded in speculative rather than empirical evidence, leading to inconclusive findings.  

ANSWERS: THANK YOU FOR YOUR COMMENT. IN THE DISCUSSION SECTION, WE FOLLOW THE PUBLISHER’S INSTRUCTIONS. WE SHOW THE EVIDENCE FROM OUR STUDY AND ENCOURAGE SCHOLARS FUR FURTHER EMPIRICAL EVIDENCE-BASED RESEARCH ON GENERATIVE AI IN EDUCATION AND SCHOOLS, AND HIGHLIGHT THE POTENTIAL OF GENERATIVE AI WHEN PROPERLY APPLIED.

  1. Many articles were presented at conferences with merits and intriguing concerns regarding generative AI in education. This study appears to generate content using a generative AI (ChatGPT) and distribute it to students in grades 4-6 without parental consent, proofreading, verification, or lesson-related connections. Then, a simple statistic was used to demonstrate the mean, standard deviation, standard error, maximum, and minimum conclusion. The manuscript contained fewer aspectual contributions (theoretical, implication, framework, suggestions for use, guideline, and future research based on this study) than typical journal articles of high-quality accuracy.

ANSWERS: THANK YOU FOR YOUR COMMENT. HOWEVER, OUR VIEWPOINTS IS DIFFERENT. THIS MANUSCRIPT IS GLOBALLY AMONG THE FIRST IF NOT THE FIRST EMPIRICAL STUDY OF THE USE OF GENERATIVE AI TO ADAPT SCHOOL LESSON MATERIAL TO PUPILS’ DIVERSE KNOWLEDGE LEVELS, INCLUDING THE RE-ADAPTATION OF THE MATERIAL AND PUPILS DURING A SCHOOL LESSON. IT CRITICALLY ANALYZES ALL RELEVANT PEER-REVIEWED SCIENTIFIC PUBLICATIONS SO FAR ABOUT THE IDEAS AND USES OF GENERATIVE AI (ESPECIALLY GPT-MODELS) IN SCHOOL EDUCATION REGARDING PUPILS’ LEARNING, INCLUDING THE RELATED RISKS. IT EXPLAINS WITH METICULOUS PRECISION HOW THE TEST LESSON WAS ORGANIZED, THE EMPIRICAL MATERIAL WAS COLLECTED AND THE USE OF GENERATIVE AI IN THIS PROCESS. THE STUDY SHOWS ITS DETAILED METHODOLOGY AND HOW RELIABLE STATISTICAL METHODS WERE USED TO ACHIEVE ITS RESULTS. FURTHERMORE, CONTRIBUTIONS TO SUSTAINABILITY THROUGH SUPPORTING SDG4 WERE INDICATED, AND FUTURE RESEARCH DIRECTIONS WERE SUGGESTED.